# In-situ constraints on the vertical distribution of global aerosol

Duncan Watson-Parris[1], Nick Schutgens[2], Carly Reddington[3], Kirsty J. Pringle[3], Dantong Liu[4], James D. Allan[5,6], Hugh Coe[5], Ken S. Carslaw[3], Philip Stier[1]

[1]Atmospheric, Oceanic and Planetary Physics, Department of Physics, University of Oxford, Oxford, UK

[2]Earth Sciences, Faculty of Science, Vrije Universiteit Amsterdam

[3]School of Earth and Environment, University of Leeds, Leeds, UK

[4]Department of Atmospheric Sciences, School of Earth Sciences, Zhejiang University, Hangzhou, Zhejiang, China

[5]Centre for Atmospheric Science, SEAES, University of Manchester, Manchester, UK

[6]National Centre for Atmospheric Science, University of Manchester, Manchester, UK

*Correspondence to*: Duncan Watson-Parris (duncan.watson-parris@physics.ox.ac.uk)

**Abstract.**

Despite ongoing efforts, the vertical distribution of aerosols globally is poorly understood. This in turn leads to large uncertainties in the contributions of the direct and indirect aerosol forcing on climate. Using the Global Aerosol Synthesis and Science Project (GASSP) database – the largest synthesised collection of in-situ aircraft measurements currently available, with more than 1000 flights from 37 campaigns from around the world – we investigate the vertical structure of sub-micron aerosols across a wide range of regions and environments. The application of this unique dataset to assess the vertical distributions of number size distribution and Cloud Condensation Nuclei (CCN) in the global aerosol-climate model ECHAM-HAM reveals that the model underestimates accumulation mode particles in the upper troposphere, especially in remote regions. The processes underlying this discrepancy are explored using different aerosol microphysical schemes and a process sensitivity analysis. These show that the biases are predominantly related to aerosol ageing and removal rather than emissions.

**Plain Language Summary**

The vertical distribution of aerosol in the atmosphere affects its ability to act as cloud condensation nuclei, and changes the amount of sunlight it absorbs or reflects. Common global measurements of aerosol including integrated properties such as Aerosol Optical Depth (AOD) provide no information about this vertical distribution. Using a global collection of in-situ aircraft measurements to compare with an aerosol-climate model (ECHAM-HAM) we explore the key processes controlling this distribution and find that wet removal (by e.g. precipitation) plays a key role.

## 1 Introduction

Atmospheric aerosol particles play a crucial role in the global energy balance by interacting with long-wave (LW) and short-wave (SW) radiation both directly and indirectly through aerosol-cloud-interactions (ACIs). The direct radiative forcing due to aerosol particles depends on their scattering and absorption properties which are primarily determined by their refractive index, size and shape. The indirect forcing in liquid clouds (and to some degree of mixed phase clouds, Heikenfeld et al.,

2019), depends on the ability of aerosol particles to act as cloud condensation nuclei (CCN), which in turn depends on the hygroscopicity and size distribution at the altitude of cloud droplet activation, which is mostly around cloud base at altitudes of 1-3 km. Hence constraining the global aerosol size distribution is a necessary (albeit insufficient) requirement for constraining both the direct and indirect aerosol forcing. In particular, the vertical distribution of aerosol, both natural and anthropogenic, can affect the magnitude of both of these effects (Samset et al. 2013; Marinescu et al. 2017).

Measurements of aerosol microphysical properties with good spatial coverage and reliability are vital for constraining the simulated aerosol properties in general circulation models (GCMs). However, currently available in-situ measurement datasets have limited global representativeness – they do not equally sample all of the relevant aerosol regimes. The Cloud-Aerosol Lidar with Orthogonal Polarization (CALIOP; Winker et al. 2009) space-borne lidar provides unique information about the vertical distribution of cloud and aerosol globally and has been used in previous model evaluation studies (Koffi et al. 2012; Koffi et al. 2016) but it is not possible to infer aerosol size information from the retrievals. Design constraints also mean that CALIOP is unable to detect background aerosol in the free troposphere because of the insufficient signal-to-noise ratio (Winker et al. 2013; Watson-Parris et al. 2018; Kacenelenbogen et al. 2011). The European Aerosol Research Lidar Network (EARLINET; Pappalardo et al. 2014) and NASA Micro-Pulse Lidar Network (MPLNET; Berkoff, Welton, and Campbell 2004) ground station networks provide continent-scale lidar measurements, and have been used in model evaluations (Ganguly et al. 2009; Satheesh, Vinoj, and Moorthy 2006) but these are unable to constrain remote aerosol conditions.

In-situ aircraft measurements provide important direct measurements of aerosol chemical composition, size distributions and radiative properties anywhere in the troposphere. These measurements have been used extensively to investigate the representation of Black Carbon (BC) in GCMs (e.g. Koch et al. 2009; Schwarz et al. 2010, Kipling et al. 2013, Reddington et al. 2013), particle number (e.g. Spracklen et al. 2007; Yu et al. 2008; Mann et al. 2014; Dunne et al. 2016), organic aerosol (Heald et al. 2011) and also aerosol size distribution (Ekman et al. 2012). However, with some notable exceptions (e.g. Clarke and Kapustin 2002), exploitation of aircraft measurements for global model evaluation has been restricted to a very small fraction of the available datasets, primarily because of the lack of easy access and a common data format. When using a selection of campaigns it is also unlikely that these accurately represent the different global aerosol regimes. The Global Aerosol Synthesis and Science Project (GASSP) dataset (Reddington et al. 2017) brings together measurements from more than 1000 flights across 37 campaigns from around the world in a consistent, synthesised format. Using this combination of aircraft datasets we are able to make more extensive evaluations of global climate models. In this paper we use GASSP to evaluate the sub-micron aerosol and CCN distribution in ECHAM-HAM - an aerosol-climate model which includes explicit treatment of the aerosol size distribution and aerosol cloud interactions.

The first focus of the paper is to illustrate the usefulness of a global aircraft dataset in evaluating aerosol in a GCM, and some of the caveats and issues in doing so. While a large collection of aircraft measurements can provide extremely valuable information about aerosol microphysical properties, there are difficulties in using such data to evaluate a GCM. For example, aircraft in-situ measurements represent a single point in space and time, whereas typical GCM output represents an average over a large (~100 km) region and often days or months (Schutgens at al. 2016b). We show that these sampling errors can be

ensured to be small compared to model errors when the measurements are averaged over time and the high-temporal resolution 4-D model fields are interpolated onto the measurement locations. The Community Intercomparison Suite (CIS) makes these interpolations straight-forward even for fields on a hybrid sigma coordinate system (Watson-Parris et al. 2016).

The second focus is on characterising the vertical distribution of aerosol particles globally by combining these measurements with a GCM. We use one-at-a-time sensitivity tests of ECHAM-HAM model simulations and employ both the M7 modal and Sectional Aerosol module for Large Scale Applications (SALSA) bin microphysics (H. Kokkola et al. 2008) schemes to explore the processes controlling these distributions. We find that ECHAM-HAM represents the aerosol size distribution well in the boundary layer, but that it appears to underestimate accumulation mode particles in the free-troposphere, which is also reflected in the CCN distribution. The wet-deposition and ageing by sulfate condensational growth are both shown to play a crucial role in these biases.

In Section 2 we describe the GASSP dataset and ECHAM-HAM model, before discussing the evaluation and sampling strategies in Section 3. We present the measurements and results from the evaluation in Section 4 and discuss their implications on constraining the global aerosol particle distribution in Section 5.

## 2 Data

### 2.1 The GASSP dataset

The GASSP dataset provides a global collection of in-situ aerosol measurements from a large number of platforms in a single self-describing data format (Reddington et al. 2017). It includes measurements from more than 1000 flights and across 37 campaigns around the world - representing remote, continental and Arctic regions in the largest collection of data of its kind. All of the campaigns that included measurements of number size distribution or CCN are included in this evaluation, as detailed in Table 1 and shown in Figure 1. The instruments providing aerosol size distributions (DMA, OPC, PILS, SMPS and FMPS) form the focus of the analysis (Section 4.1), while the CCN Counter (CCNC) provides measurements of CCN used in Section 4.4, and Sulfate measurements from the AMS instruments are used in Appendix C.

**Table 1: Details of the campaigns and instruments included in the GASSP database that were used in this analysis. The environmental conditions summarise the prevailing conditions during the measurements, see associated references for more details.**

| Campaign label | Aircraft | Season | Environmental conditions | Instrumentation |
|---|---|---|---|---|
| ACCACIA (Lloyd et al. 2015; Young et al. 2016) | FAAM BAe-146 | Spring 2013 | - | AMS[1] |

---

[1] Aerodyne Aerosol Mass Spectrometer C-TOF (Canagaratna et al. 2007; Drewnick et al., 2005)

| | | | | |
|---|---|---|---|---|
| ACE1 (Clarke et al. 1998) | NCAR C-130 | Summer (SH) 1995 | Marine, clean | DMA[2] + OPC[3], CCNC[4], PILS[5] |
| ACEASIA (McNaughton et al. 2004; Howell et al. 2006) | NCAR C-130 | Spring (NH) 2001 | Asian outflow of dust, soot (biomass burning), anthropogenic aerosol, anthropogenically modified atmosphere, polluted marine | DMA[6] + OPC[3] |
| AEGEAN-GAME (Bezantakos et al. 2013) | FAAM BAe-146 | Summer 2011 | Marine boundary layer, Prevailing northern winds (the Etesians), polluted airmasses | SMPS[7], AMS[1] |
| APPRAISE (Crosier et al. 2011) | FAAM BAe-146 | Winter 2009 | Missions typically involved flight legs above and below cloud layers to characterise aerosol in the vicinity of the clouds, and flight legs within cloud to characterise cloud properties and attempt to measure cloud particle residuals | CCNC[8] |
| ARCPAC2008 (Brock et al. 2011) | NOAA WP-3D | Spring 2008 | Arctic haze, Transport Asian pollution, High-latitude biomass burning | CCNC[8], AMS[1] |
| ARCTAS (Jacob et al. 2010) | NASA P-3B + NASA DC-8 | Spring, Summer 2008 | Arctic haze, Boreal forest fires / biomass burning, Long-range transport of pollution to the Arctic | DMA[9] + OPC[10] + CCNC[8], AMS[11] |
| BORTAS (Palmer et al. 2013) | FAAM BAe-146 | Summer 2011 | Boreal forest fire, biomass burning | SMPS[7], AMS[1] |

[2] Radial Differential Mobility Analyzer (RDMA) is a small custom-built disc DMA (Zhang et al. 1995) with particle sizing set at 0.01-0.25 um using a TST Model 3010 with 22°C saturator temperature difference for lowered detection limit and with thermal analysis similar to the OPC

[3] Laser optical particle counter (OPC) (Particle Measurement Systems LAS-X, Boulder, Colorado with customized electronics) effectively sizes particles between 0.100 and 14 um with a resolution of 112 logarithmically spaced channels per decade (Clarke 1991)

[4] Desert Research Institute (DRI) instantaneous CCN spectrometer (Hudson 1989). Parallel plate thermal gradient diffusion cloud chamber with streamwise supersaturation gradient (each plate is divided into eight temperature controlled zones).

[5] The Georgia Institute of Technology particle-into-liquid sampling (PILS) system

[6] NCAR radial differential mobility analyzer system (NCAR rDMA) measured the size and number of particles between 0.007 and 0.150 um with a resolution of 54 channels per (Russell et al. 1996)

[7] Bespoke Scanning Mobility Particle Sizer (using TSI 3081 DMA, Custom 3786-LP WCPC, 4143 Flowmeter)

[8] Droplet Measurement Technologies Stream-wise Thermal Gradient Continuous-Flow CCN Counter (Roberts and Nenes, 2005; Lance et al., 2006)

[9] Two custom DMA's (TDMA 0.01-0.20 um, LDMA 0.010-0.50 um mobility diameter)

[10] Optical particle counter (0.15-8.0 um optical diameter)

[11] High-resolution time-of-flight aerosol mass spectrometer (HR-ToF-AMS) (Canagaratna et al. 2007; DeCarlo et al. 2006).

| | | | | |
|---|---|---|---|---|
| CALNEX (Ryerson et al. 2013) | NOAA WP-3D | Spring 2010 | Polluted, Urban plumes/emissions, Ship plumes/emissions, Rural | CCNC[8], AMS[1] |
| COPE (Leon et al. 2016) | FAAM BAe-146 | Summer 2013 | - | SMPS[7], AMS[1] |
| DC3 (Barth et al. 2015) | NASA DC-8 | Spring, Summer 2012 | Midlatitude continental convective clouds, Convective storm inflow, Convective transport of fresh emissions | CCNC[8], AMS[11] |
| EM25 | FAAM BAe-146 | Summer 2009 | Urban polluted, direct sampling of emissions from traffic on M25 motorway, in-plume sampling | CCNC[8] |
| EUCAARI (Hamburger et al. 2011) | FAAM BAe-146 | Spring 2008 | Polluted, high-pressure system, ageing pollution | CCNC[8], AMS[11] |
| GoAmazon (Martin et al. 2016) | ARM Aerial Facility (AAF) Gulfstream-1 (G-1) | Dry season 2014 | Pollutant outflow from a tropical megacity, Urban plume sampling, Evolution of properties along the Manaus plume, persistent easterly winds, sampling pristine and polluted air masses, sampling changes to gases and particles within detrainment levels of shallow cumulus clouds | CCNC[8] |
| INDOEX (Twohy et al. 2001, Clarke et al. 2002) | NCAR C-130 | NH Winter 1999 | Polluted air from Indian subcontinent blown by low-level winds over Arabian Sea and south to the equator. | OPC[3], CCNC[4] |
| INTEX-A (Clarke et al. 2007) | NASA DC8 | Summer 2004 | Inflow and outflow of pollution over North America, continental pollution plumes, biomass burning plumes | DMA + OPC[3], PILS[12] |
| INTEX-B (Singh et al. 2009) | NCAR C-130 | Spring 2006 | Polluted, marine boundary layer | CCNC[8], AMS[11] |

---

[12] Particle into Liquid Sampler coupled to ICs one for the anions and one for the cations

| | | | | |
|---|---|---|---|---|
| MIRAGE (DeCarlo et al. 2008)[13] | NCAR C-130 | Spring 2006 | Megacity urban pollution, coastal regions, Regional influence of Mexico City significant relative to other regions. | FMPS [14], CCNC[8], AMS[11] |
| PASE (Clarke et al. 2013) | NCAR C-130 | None 2007 | Focus on chemistry and physics (primarily of sulfur) in a cloud free convective boundary layer of Pacific Ocean. | DMA[6] + OPC[3], CCNC[4], AMS[11] |
| PEMTropicsA (Hoell et al. 1999) | NASA P-3B | NH Summer, NH Autumn, Southern-tropical dry season 1996 | Marine airmasses, Campaign period is a time of enhanced biomass burning in the southern hemisphere and this influence was evident in the data. | DMA[2] + OPC[3] |
| PEMTropicsB (Clarke et al. 2001) | NASA P-3B | NH Winter, NH Spring, Southern-tropical wet season 1999 | Marine airmasses, tropical marine boundary layer, La Nina event, Continental pollution outflow, Long-range transport of dust and combustion aerosol plumes from Asian continent [Clarke et al., JGR, 2001] | DMA[2] + OPC[3] |
| RONOCO (Walker et al. 2015) | FAAM BAe-146 | Summer 2010 | The majority of the flying took place at night, with occasional flights beginning or ending in daylight hours to study chemical behaviour at dusk and dawn, urban pollution | CCNC[8], AMS[11] |
| SEAC4RS (Toon et al. 2016) | NASA DC-8 | Summer, Autumn 2013 | polluted airmasses, sampling wildfire / biomass burning plumes in western USA e.g. the Rim Fire, sampling natural summertime emissions of isoprene from forests in the Southeast USA, deep convective outflow and clouds | CCNC[8], AMS[11] |
| TEXAQS2006 (Parrish et al., 2009; Asa-Awuku et al. 2011) | NOAA WP-3D | Summer, Autumn 2006 | polluted, near the end of the summer photo-chemical ozone production season, majority of flights occurred in Houston urban and industrial ship-channel area, aircraft sampled urban outflow from Dallas, Other major targets | CCNC[8], AMS[11] |

[13] Note that the MIRAGE campaign (more specifically MIRAGE-Mex) was designed to sample the heavily polluted air downwind of Mexico City, Mexico and as such introduced a large sampling bias when compared with the GCM area average (Tie et al. 2009). It was not included in the subsequent analysis.

[14] TSI Fast Mobility Particle Sizer (FMPS) Spectrometer

| | | | included power plant and various industrial plumes in east Texas, Biomass-burning plumes were sampled on a few occasions | |
|---|---|---|---|---|
| TROMPEX (Andrews et al. 2013) | FAAM BAe-146 | Summer, Autumn 2009 | Marine airmasses | CCNC[8] |
| VOCALS (Wood et al. 2011) | NCAR C-130 | NH Autumn, Southern-tropical dry season 2008 | Marine airmasses, Continental pollution outflow, Large stratocumulus cloud deck, | DMA[6] + OPC[3], CCNC[15], AMS[11] |
| VOCALS (Wood et al. 2011) | FAAM BAe-146 | As above | As above | SMPS[7], CCNC[8], AMS[11] |

---

[15] University of Wyoming CCN instrument consists of a static thermal-gradient chamber and an optical detection system (Snider et al. 2006)

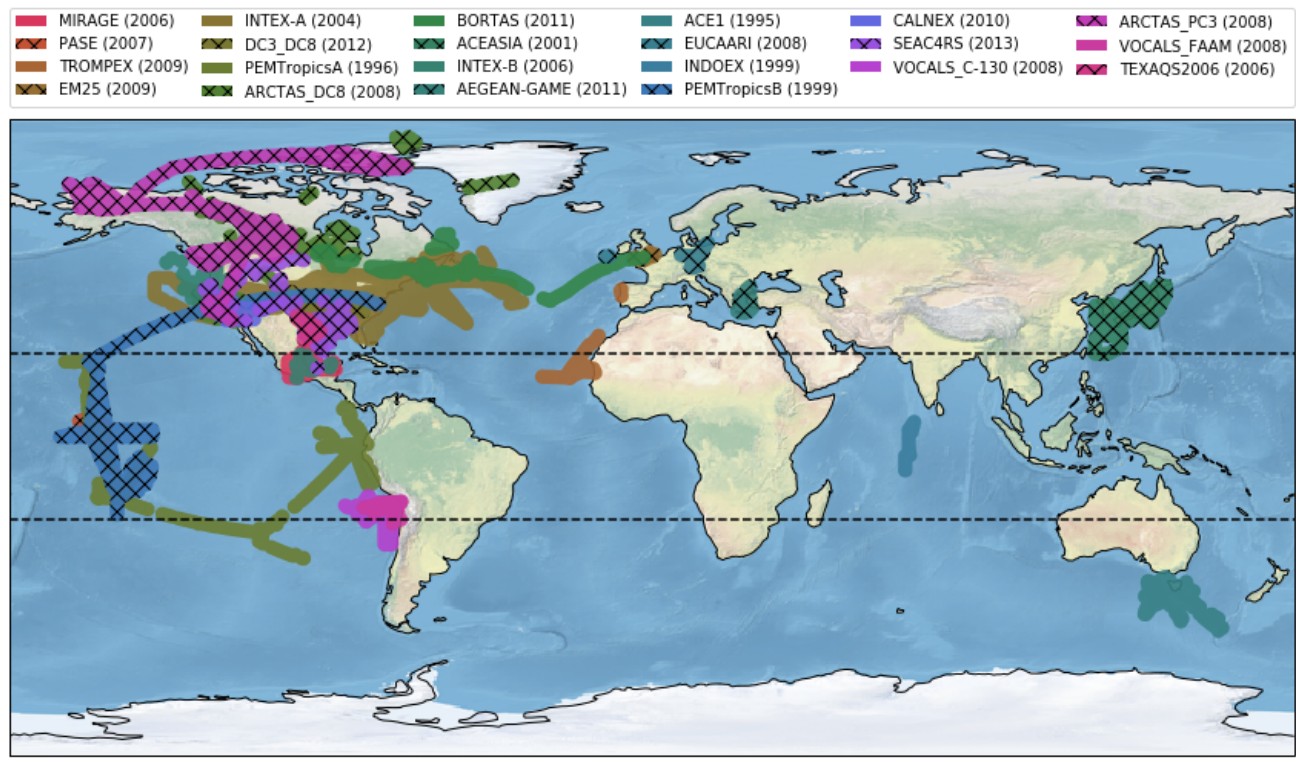

**Figure 1: Spatial coverage of flight campaigns included in the GASSP database which include number size distribution or CCN measurements**

## 2.2 Model description

In this study we use the ECHAM-HAMMOZ model as an example of a modern, well-characterized global aerosol-climate model in order to demonstrate the value of the GASSP dataset. While other models will likely show different behaviours and biases it is hoped this initial evaluation will inform an extended analysis across a large number of models. The recently released ECHAM6.3-HAM2.3 version is used, which includes improved sea-salt and dust emission parameterisations and is described and evaluated in detail by Tegen et al. (2019) and Neubauer et al. (2019).

Briefly, the ECHAM6 atmospheric general circulation model (Stevens, et al. 2013), which is developed by the Max Planck Institute for Meteorology in Hamburg, Germany, utilizes a spectral transform dynamical core and a semi-Lagrangian tracer transport scheme in flux form (Lin and Rood, 1996). Convection is parameterized via the mass-flux schemes by Tiedtke (1989) and Nordeng (1994) and sub-grid-scale stratiform clouds use the scheme of Sundqvist et al. (1989). While the base model uses a 1-moment cloud microphysics scheme ECHAM-HAM uses a 2-moment cloud microphysics scheme (Lohmann et al. 2007;

Lohmann and Hoose 2009; Lohmann and Neubauer 2018)

The microphysical aerosol model HAM (Stier et al., 2005; Zhang et al. 2012; Tegen et al., 2019) computes the evolution of an external mixture of internally-mixed log-normal aerosol modes, considering the species sulfate, black carbon (BC), organic

carbon (OC), sea salt, and mineral dust. Coupled to ECHAM, the evolution of the log-normal modes, represented by aerosol mass and numbers, is computed taking into account physical and chemical particle processes. The microphysical core M7 (Vignati et al., 2004) calculates coagulation among the modes and the condensation of gas-phase sulfuric acid on the existing aerosol population. Aerosol particles are removed by sedimentation and dry and wet deposition. Gravitational sedimentation of particles is calculated based on their median size using the Stokes settling velocity (Seinfeld and Pandis, 1998), with a correction factor according to Slinn and Slinn (1980). Removal of aerosol particles from the lowest model layer by turbulence depends on the characteristics of the underlying surface (Zhang et al., 2012). The aerosol dry deposition flux is computed as the product of tracer concentration, air density, and deposition velocity, depending on the aerodynamic and surface resistances for each surface type considered by ECHAM6.3, and subsequently added up for the fractional surface areas. For wet deposition the in-cloud scavenging scheme from Croft et al. (2010), dependent on the wet particle size, is used. The in-cloud scavenging scheme takes into account scavenging by droplet activation and impaction scavenging in different cloud types, distinguishing between stratiform and convective clouds and warm, cold, and mixed-phase clouds. Below clouds particles are scavenged by rain and snow using a size-dependent below-cloud scavenging scheme (Croft et al., 2009). Scavenged particles can also be resuspended by the evaporation of precipitation (Stier et al, 2005). In turn, the effects of aerosols on clouds and radiation are computed prognostically in the coupled ECHAM–HAM. The relative importance of the individual aerosol processes in ECHAM–HAM has been evaluated by Schutgens et al. (2014) and informs the choice of parameters used for the sensitivity analysis in this work.

In this work the model is run at T63 resolution with 31 vertical levels and nudged to ERA-Interim reanalysis for 2008 (Dee et al. 2011), using the ACCMIP interpolated emission dataset (Lamarque, Bond, and Eyring 2010). The use of a single model year to compare against the multi-year GASSP dataset and its implications for the evaluation are discussed in Section 3.

In order to explore the uncertainty in the vertical structure of the aerosol size distribution we perform a set of sensitivity simulations in which key parameters were scaled up and down one at a time. There are dozens of parameters and processes which affect the aerosol number at any given location (e.g. Lee et al. 2013) but this analysis aims to cover the main processes which would be expected to affect the relative biases in size distribution discussed in Section 4.1, particularly focussed around aerosol growth, removal and vertical transport. There are some processes which might be expected to affect these biases which are not represented in the model at all such as prognostic in-cloud aerosol processing (Hoose et al. 2008); aerosol removal by photolysis (Hodzic et al., 2015) and interactive Secondary Organic Aerosol (SOA) formation (e.g. Heald et al. 2011). For example, the use of prescribed SOAs has been shown to result in a lower SOA burden compared to online calculation (Tegen et al. 2019). This has also been shown to have a large effect on the vertical distribution of organic aerosol in other models (e.g. Shrivastava et al. 2015; Tsigaridis et al. 2014), although the exact impact on the vertically resolved aerosol size-distribution considered in this work is not clear. While the contribution of these processes are not explored further here, such structural uncertainties will be the focus of a future multi-model experiment described in Section 5.

Having determined the processes to analyse, we apply a simple high/low perturbation over their likely range of uncertainty in order to determine their relative contribution to any model-measurement differences, as outlined in Table 2 and described in

detail below. One-at-a-time sensitivity tests neglect the important effects of combinations of parameter perturbations that are captured by Latin hypercube perturbed parameter ensemble studies (e.g. Lee et al. 2013; Regayre et al. 2018). However, they allow an assessment of how individual process parameter uncertainties contribute to the vertical profile of the aerosol size distribution and can be used as a screening test to determine important model processes for further analysis.

**Table 2: Outline of the sensitivity experiments performed. The scale factors refer to the high and low multiplicative factors applied to the relevant model parameter. See text for details.**

| Parameter | Description of scaled quantity | Scale factor |
|---|---|---|
| Condensational ageing | The number of layers of SO4 for a particle to be 'coated' | 0.3 - 5.0 |
| Wet deposition | In- and below-cloud wet deposition fluxes | 0.5 - 2.0 |
| Vertical flux in convection | Convective tracer entrainment | 0.1 - 10.0 |
| Coagulation | Probability of inter- and intra-mode coagulation | 0.5 - 2.0 |
| Dry deposition (acc.) | Dry deposition of accumulation mode aerosol | 0.1 - 10.0 |
| Dry deposition (Ait.) | Dry deposition of Aitken mode aerosol | 0.5 - 2.0 |

**Condensational ageing.** In the default HAM setup a single monolayer of sulfate is assumed to be required to transfer insoluble particles to the corresponding soluble/mixed particle mode following (Vignati, Wilson, and Stier 2004). There is considerable uncertainty in this simple approximation. We therefore vary the number of monolayers required from 0.3 to 5, matching the ranges used by (Lee et al. 2013).

**Wet deposition.** HAM2 includes wet deposition removal of aerosol via in-cloud nucleation and impaction scavenging as well as below-cloud impaction scavenging by rain and snow. In-cloud nucleation is the most important of these mechanisms and, because it primarily occurs at the top of the boundary layer, it has a large effect on the vertical distribution of aerosol (e.g. Kipling et al. 2016; Mahmood 2016). Here we scale the total in-cloud and below-cloud mixing-ratio removal tendencies in each grid cell by a constant factor. As one of the primary aerosol removal mechanisms globally, the aerosol burdens are very sensitive to this scaling; so while there are large uncertainties in the precipitation and scavenging rates, the range of scalings used is smaller than in the other perturbations. Initial scaling values of 10, 5 and even 3 led to implausible aerosol burden globally.

**Vertical flux in convection.** Convection is one of the dominant mechanisms for transporting aerosol and trace gases from the boundary layer into the free troposphere globally (Park and Allen 2015). There are large uncertainties in the aerosol entrainment and detrainment rates for convective clouds. Here we scale the total convective tracer mass flux in each grid-cell to sample this uncertainty. The large range in scale factors was chosen to reflect the large uncertainties in the fluxes, and due to the relative insensitivity of the aerosol to this parameter.

**Coagulation.** The inter- and intra-modal components of the standard coagulation kernel within the M7 aerosol scheme can be scaled independently to represent uncertainty in the assumptions used to calculate it - such as using only the median mode

diameter in calculating the terms, and uncertainties in the effects of turbulence and electrostatics. In this work, since we are interested only in the broad uncertainties, we scale the whole kernel by the same scale factor.

**Aerosol dry deposition.** Lee et al. (2013) showed that uncertainties in the dry deposition process provided the largest contribution to the uncertainty in CCN globally in the HadGEM-GLOMAP GCM. Here we scale the dry deposition velocities for the Aitken and accumulation modes across the same range as in their study.

## 3 Evaluation strategy

The GASSP aircraft database provides valuable measurements with which to constrain global climate models. However, these near-instantaneous point measurements represent something quite different to typical model output fields which are often temporal averages of a grid-cell which itself represents some (usually undefined) average over a large spatial region, typically ~100 km in extent. The question of how to compare these two datasets consistently is the subject of this section.

Schutgens et. al (2016a) show the importance of collocating measurements with high temporal resolution model fields in order to reduce the large temporal sampling artefacts which would otherwise be present. This has also been noted in previous model evaluation work using aircraft data (Ekman et al. 2012). Further work (Schutgens et al. 2016b) showed the importance of averaging these collocated measurements over as long a period as possible in order to remove spatial sampling biases. While higher spatial-resolution models would reduce this particular form of sampling bias, they would face the associated problem of small transport differences leading to large biases (Fast et al. 2016). Some campaigns include sampling biases by design, due to the particular objectives of the mission. For example, the MIRAGE campaign was flown to specifically measure the pollution down-wind of Mexico City and hence over represents the mean aerosol loading in the (wider) region. As noted in Table 1 this campaign was not included in the subsequent analysis. Some recent flight campaigns, such as VOCALS (Wood et al. 2011) and ORACLES (Zuidema et al. 2016), fly routine tracks several times during the campaign specifically to build up representative spatial statistics for comparison with models, however most historic campaigns have not.

A further complication in the use of the combined measurements from a variety of campaigns, aircraft and even instruments is that the measurements themselves will have different sampling rates and systematic biases (for example due to the use of different inlets). Another concern is the different inlets and piping used to bring the sampled air inside the aircraft and to the instrumentation. Because these biases will generally be uncorrelated across campaigns, we assume that the large number of campaigns used will remove any systematic bias in the reported average size distributions.

As discussed in Section 2.3 the model is run for a single year rather than the full observational period. Although the GASSP data spans many years, the inter-annual variability in aerosol burden, away from the main biomass burning regions, is small (Li et al. 2013). Further, the three oldest campaigns (ACE-1 and PEM-Topics A/B) sample remote ocean environments where the effect of any trends in anthropogenic emissions are expected to be small. Nevertheless, inter-annual variability and trends in meteorology, emissions and removal (through e.g. precipitation) will introduce some uncertainty in our analysis.

In order to remove the most high-frequency variability in the measurements (which we would not expect the GCM to reproduce), to bring the measurements onto a common temporal sampling, and to provide at least some temporal aggregation we down-sample the measurements to 2-minute averages. Typical aircraft in the GASSP database, such as the NOAA P3-B and the FAAM BAE-146, have cruise speeds of 600-800 km/h, so this averaging corresponds to a distance of 10-15km. Detailed investigations of spatial variability of aircraft aerosol measurements in the ARCTAS campaign (Shinozuka and Redemann 2011) show that this length scale will average out local emission sources while still maintaining the long-range variability which we hope the GCM to reproduce.

Using CIS (www.cistools.net: Watson-Parris et al. 2016) to linearly interpolate the model fields of interest onto these temporally averaged measurements we should minimise the associated sampling errors. The question remains however what model output frequency is required. The storage requirements for the 3-D model fields we wish to interpolate quickly become inhibiting for daily and sub-daily model output.

Figure 2 shows the model CCN at 1% supersaturation output at different temporal frequencies interpolated onto the observation points over North America compared against the same model values output from an online flight-track simulator. This simulator provides the highest possible output frequency by interpolating arbitrary model fields onto a set of latitude/longitude/pressure points at every model output time-step. The reduced correlation (< 0.7), high normalized bias (< -0.15), decreased variability and large Root Mean Square Error (RMSE) introduced by interpolating monthly model output fields are clearly seen. Similar characteristics can be seen for the points over the South-East Pacific, however in this case the bias introduced is positive. Similar results can be obtained for other model fields all over the globe. Although the flight track simulator provides a powerful diagnostic capability, we use model data interpolated from 3-hourly output fields for the results presented in Section 4 as a compromise between the introduced sampling bias and convenience in analysis. As shown in Figure 2 the bias and RMSE introduced are negligible.

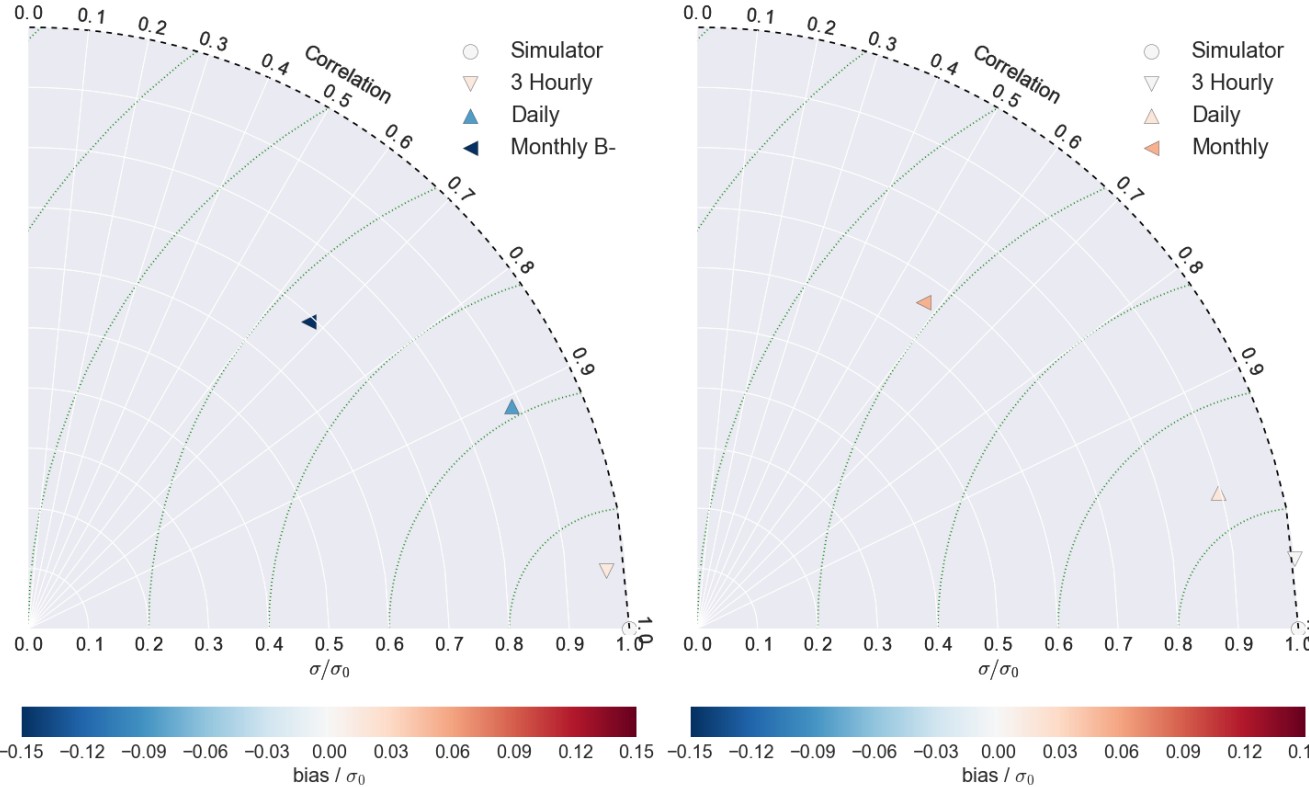

**Figure 2: Taylor diagram showing the RMS error and reduction in correlation and variability introduced by comparing instantaneous model output (using a flight track simulator) with interpolations from different temporal aggregations of the same model data. All datasets are linearly interpolated from the same underlying 4-D model field (CCN at 1% supersaturation) and compared with an online interpolation. (a) shows points taken over North America, (b) shows points over the South East Pacific.**

## 4 Results

### 4.1 Aerosol size distribution

The aerosol size distribution can be characterised in several ways, for example by aerosol number, surface area or volume. The Aitken and accumulation modes are most important for constraining the indirect effect, so we focus on the aerosol number size distribution (NSD):

$n_N^e(\ln D_P)d\ln D_p$ = number of (dry) particles per unit volume of (ambient) air in the size range $\ln D_P$ to $\ln D_P + d\ln D_P$

Figure 3 shows the median measured aerosol NSD from the GASSP database for all flights which included the relevant measurements, interpolated onto common aerosol diameters and binned into 0.5km vertical bins. (Figure 13 Appendix A shows the number of observations in each altitude bin). A number of interesting features are apparent. The strong anthropogenic sources in the northern extra-tropics are clearly seen in the larger number of accumulation mode aerosol in the boundary layer.

The tropics show similar number of aerosol in the free troposphere as the northern extra-tropics, apart from a clear increase in Aitken mode aerosol at 8 km, resulting from the growth of the significant number of nucleation mode aerosol in the Upper Troposphere, Lower Stratosphere (UTLS - Clarke and Kapustin 2002; Stier et al 2005). The aerosol distribution in the southern extra-tropics is noticeably smaller than in the tropics or northern extra-tropics with more aerosol residing in the Aitken mode, partly due to the lower tropopause height at mid-latitudes affecting the UTLS nucleation mode aerosol described above. The lower number of accumulation mode aerosol in the Southern Hemisphere has been observed before (e.g. Minikin et al. 2003) and is attributed to the lack of anthropogenic aerosol and gaseous pre-cursor sources in the Southern Hemisphere.

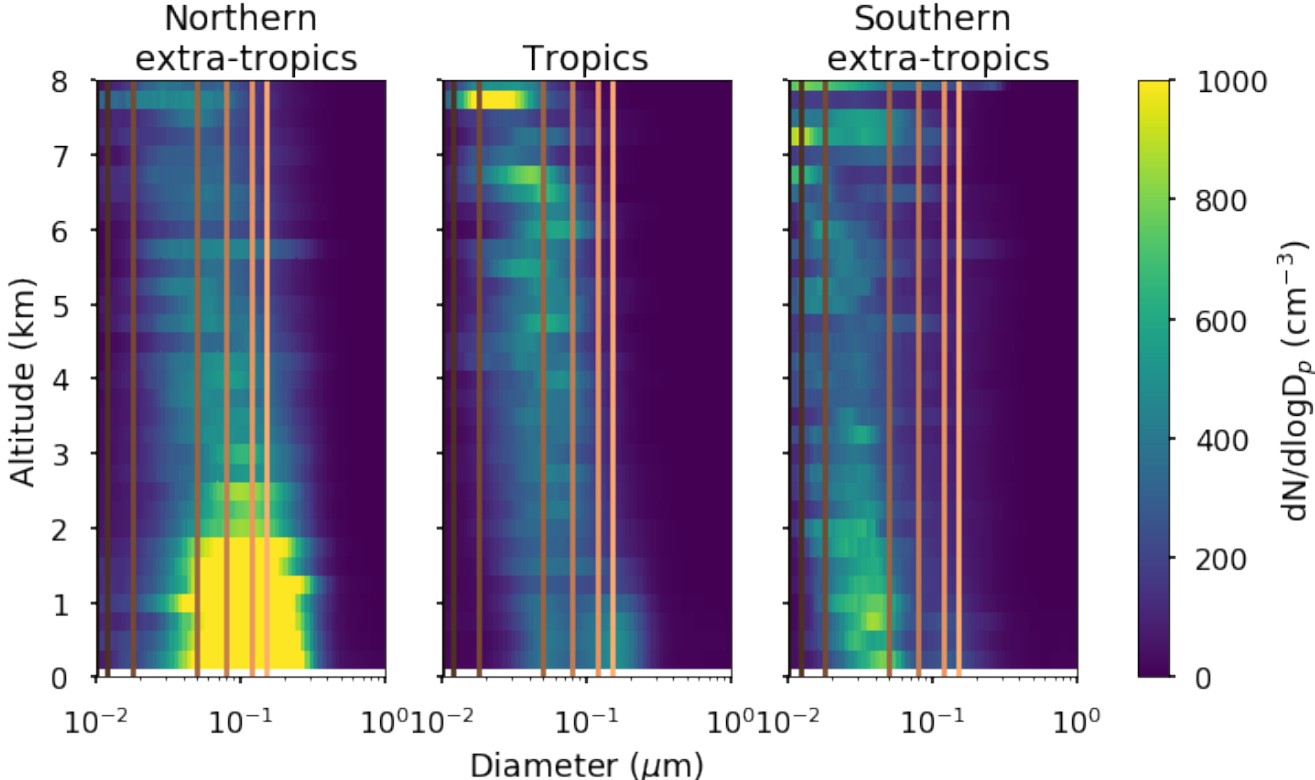

Figure 3: The median aerosol number size distribution observed at each altitude across all flights in the Northern extra-tropics (a), Tropics (b) and Southern extra-tropics (c). The copper lines represent the sizes from which the integrated numbers in e.g. Figure 4 are calculated.

For comparison between the modelled (modal) and observed (binned) aerosol distribution it is useful to reduce this distribution to a single number representing the integrated number above some lower threshold:

$$N_S = \int_S^\infty n_N(D_P)dD_P$$

While $D_P$ will often be plotted in units of $\mu m$, in this paper $N_S$ is always an integrated number above diameter $S$ in nm. We can interpolate the model mode number and radius fields on to the measurements and calculate $N_S$ across a range of sizes for each point. While the integrated number concentrations at smaller size cut-offs will include the number of larger sized

particles the smallest particles will dominate the number. Figure 4a and b shows the global average of these collocated, integrated numbers as a function of altitude. In this figure we show the median and inter-quartile range of the global values in each altitude bin as this better represents the (log-skewed) distributions. We also then plot the fractional biases, defined as

$$FB = \frac{Model - Obs}{(Model + Obs)/2},$$ as shown in Figure 4c. Profiles for each campaign and size cut-off are shown in Figure 18 (Appendix C).

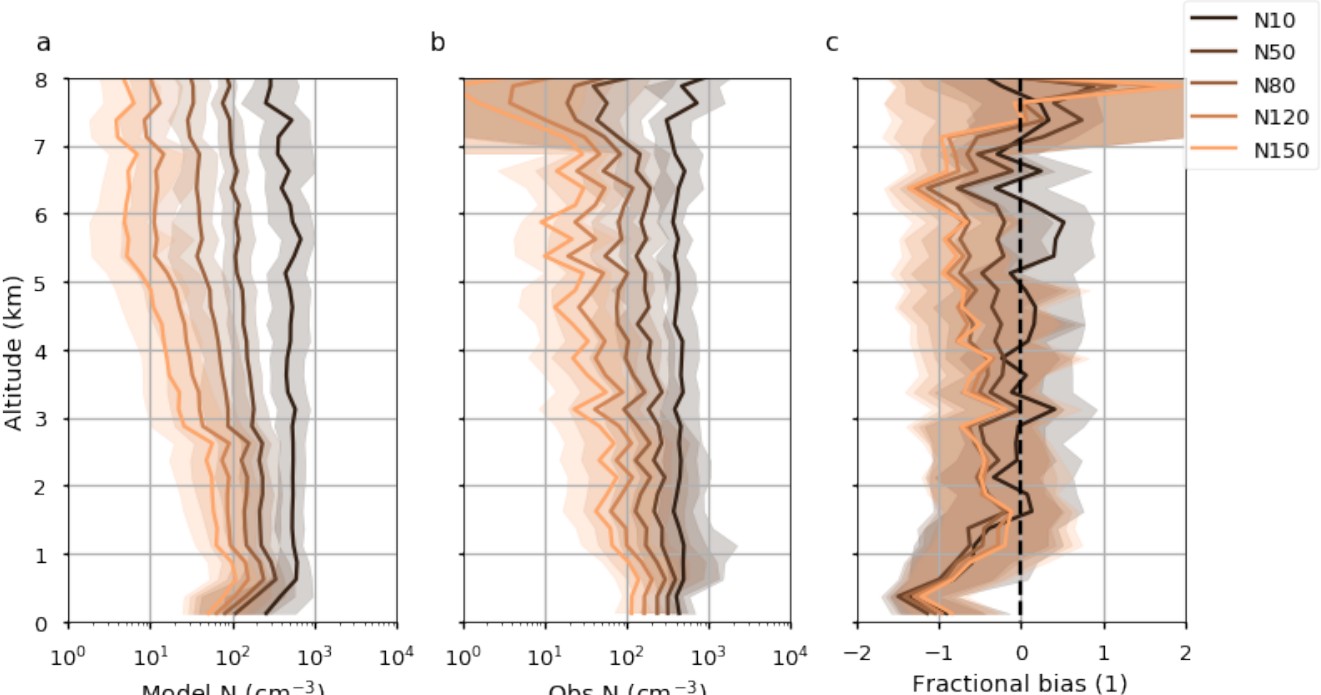

**Figure 4: The vertical profile of integrated (a) model and (b) observed aerosol number at different size cut-offs. The lines represent the median over all points in that altitude bin, while the shading shows the inter-quartile range. (c) shows the median fractional bias.**

The model performs best in the lower free troposphere with a near-zero bias around 1-3km, but it underestimates number
10 concentrations in all aerosol size ranges in the planetary boundary layer (PBL). In the upper free troposphere (FT), above 4km, the model reproduces the number of particles smaller than around 10 nm very well, but there is a clear low bias of up to 100% in the number of larger (sub-micron) particles.

It should be noted that with the DMA-based instruments there is a potential for uncertainties associated with the assumed particle charging model. As part of their inversions, these instruments must assume a probability that particles of a given size
15 achieve the specified states when subjected to the bipolar charge field at the instrument's inlet (Liu and Pui, 1974). The most common method used is the parameterisation of Wiedensohler (1988), which has proven to be robust in most applications (Wiedensohler et al., 2012). More recently, fundamental modelling studies have investigated how much this function depends on conditions such as particle composition, temperature and pressure and have suggested these effects may be important in

some situations (López-Yglesias and Flagan, 2013). In principle, if the charging function were to vary with pressure, this could be responsible for systematic artefacts in the vertical profiles of particle concentrations presented here. Leppä et al. (2017) presented a case study at 10 km altitude and suggested that the number concentrations of particles greater than 10 nm diameter would be under-reported by between 5 and 33% depending on the ambient size distribution and other technical details such as the polarity of the instrument. However, at the time of writing, we are not in a position to use this result as the basis for a correction for our data because the altitude dependency case studies of López-Yglesias and Flagan (2013) and Leppä et al. (2017) varied both temperature and pressure simultaneously according to typical ambient conditions. In contrast, the instruments whose data are being used here charged the aerosols at aircraft cabin temperature rather than ambient, so the actual effect that pressure variations may be having on the data is currently uncertain. Because applying a systematic correction would be both technically challenging and computationally expensive, this is deemed outside the scope of this work, however in the event that a generalised correction method be developed in the future, this issue should be revisited. Taken at face value however, this would mean our measured data of particles larger than 10nm would be at worst biased *low* a few tens of percent at altitude, which would in turn only make the reported model biases more significant. Note also that this uncertainty relates to how many particles get charged (and hence counted), but once charged the size of the aerosol can still be accurately determined, hence this issue only affects counting and not the sizing of particles. The effects of variations in pressure and temperature on DMA sizing are already well-established and accounted for (Knutson and Whitby, 1975).

In order to understand the source of these global model biases we can split the data into measurements made over land or ocean, roughly analogous to near / far from major sources respectively, in order to understand the role of emissions and removal in the model bias. Figure 5 shows the fractional bias in modelled aerosol number over land and ocean. The vertical profile of the bias over land shows the model consistently underestimates aerosol across all sizes in these regions. This bias is largest in the boundary layer where biases due to the sampling of local emissions sources not resolved by the coarse model resolution are likely to be dominant. The bias in the smaller particles improves with altitude and is near-zero above 6 km. The larger particles however show the same bias as in the global mean above 4km. The ocean profiles generally show a much better agreement with the measurements throughout the troposphere, although the bias in the number concentration of large particles in the free troposphere remains. In these generally more remote regions the model recreates the overall aerosol number well, but is underestimating the number of larger particles aloft - suggesting over-efficient removal of these particles, or insufficient growth. Figure 5 also shows a strong low bias in the near surface aerosol number over the ocean, for all aerosol sizes. This is (at least partly) due to insufficient SO4 in the ocean boundary layer, as shown in Figure 15, presumably due to insufficient DMS emission. Insufficient emissions of marine organic aerosols and aerosol precursors could also contribute to this bias.

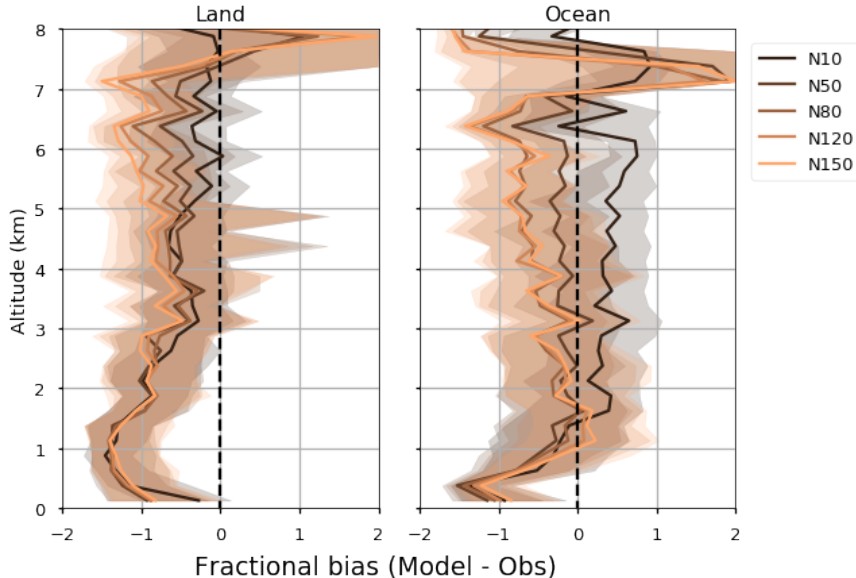

**Figure 5: The vertical profile of fractional bias in modelled aerosol number at different size cut-offs for Land (a) and Ocean (b) measurements.**

It is instructive to stratify by latitude. Figure 6 shows the fractional bias profiles for flights in the Tropics and Northern and Southern extratropics. The Northern extratropical profiles show similar biases as the over-land biases shown in Figure 5 since these include many of the same flights. Similarly, the tropical profiles are mostly over Ocean. However, the most remote dataset in the Southern extratropics (ACE1) has the strongest bias in the number of large particles.

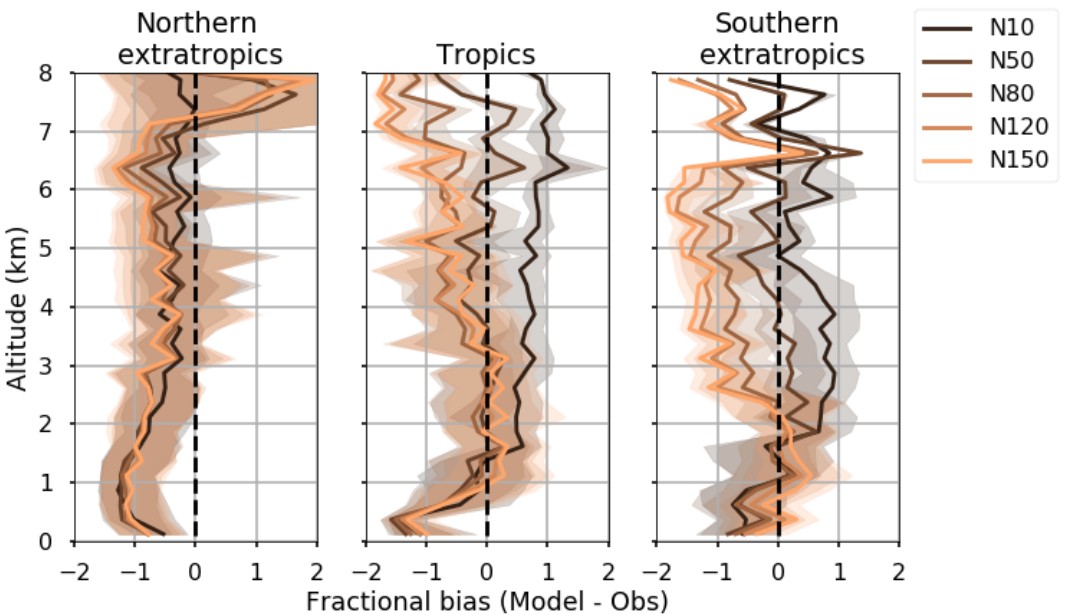

**Figure 6: The vertical profile of fractional bias in modelled aerosol number at different size cut-offs for measurements in the Northern extratropics (a), the Tropics (b), and the Southern extratropics (c).**

## 4.2 Sensitivity tests

**Condensational ageing**

In HAM-M7 condensational ageing is the primary mechanism by which aerosol (mass and number) is transferred from the insoluble Aitken and accumulation modes into their soluble equivalents (Schutgens and Stier 2014) where they become available for removal by nucleation scavenging. The current assumption in this model is to require one-monolayer of sulfate condensed on a particle to transfer it from the insoluble to soluble modes. Figure 7 shows profiles of fractional bias in each of the latitudinal ranges shown in Figure 6 but with a reduced and increased amount of sulfate required, leading to faster and slower condensational ageing respectively. Changing the condensational ageing rate between the values chosen for this study has a minimal effect in the tropical and southern extratropical regions. However, the slower condensational ageing profiles show a reduced negative bias in the number of larger particles in the northern extratropics (presumably by reducing their removal through scavenging), at the expense of an increase in the negative bias for smaller particles. Nearer the large anthropogenic sulfate sources in the Northern Hemisphere, ageing timescales are much shorter (Schutgens and Stier 2014) and hence more sensitive to these perturbations. It should be noted that ECHAM-HAM does not simulate the effects of nitrate which would be expected to contribute to the ageing of aerosols.

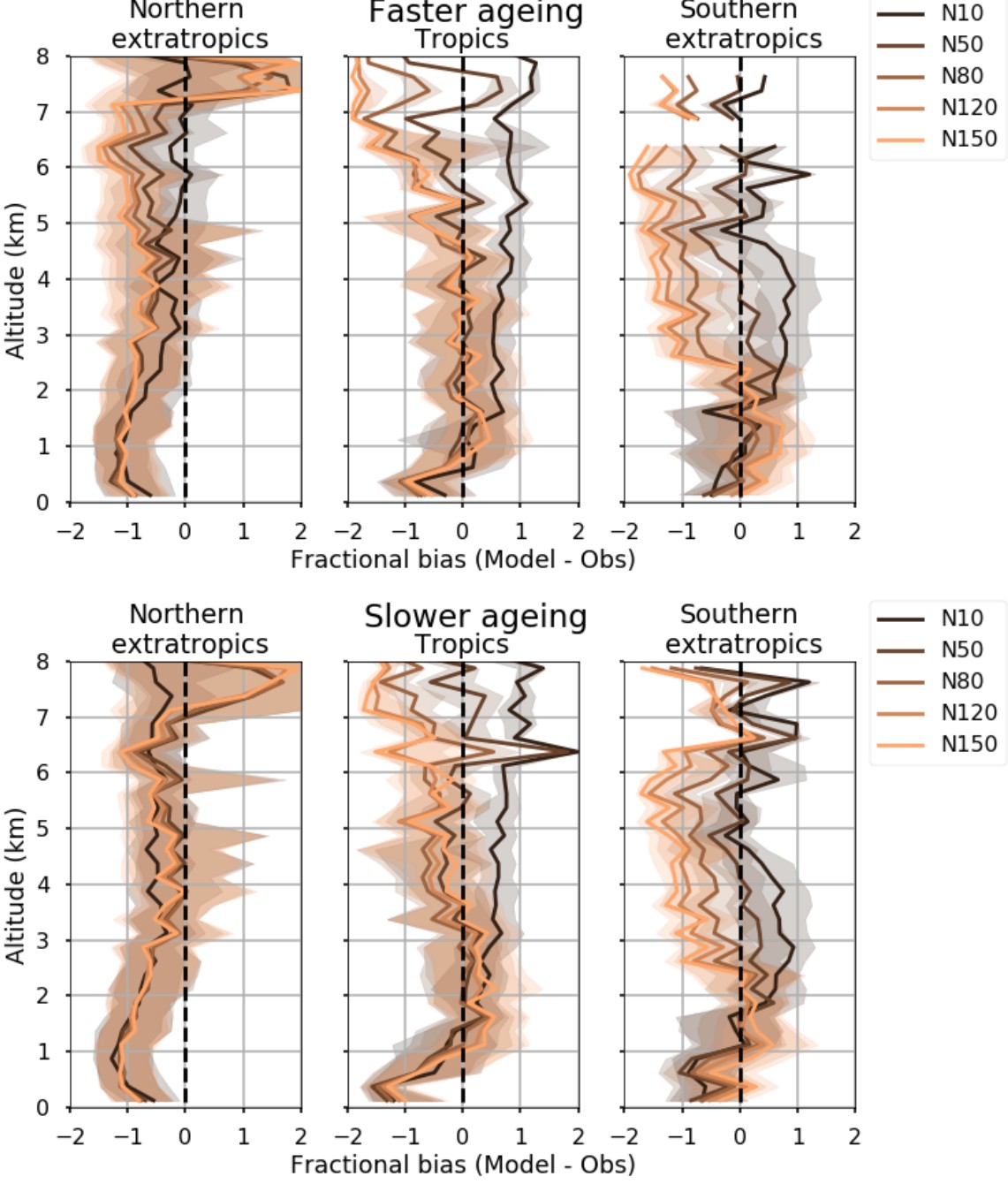

**Figure 7: The vertical profile of fractional bias in modelled aerosol number at different size cut-offs for measurements in the Northern extratropics (a), the Tropics (b), and the Southern extratropics (c) with a faster (0.3 mono-layers required) and slower (5 mono-layers) condensational ageing rate.**

**Wet deposition**

Wet deposition is the primary removal mechanism of aerosol (Textor et al., 2006) as well as in ECHAM-HAM (Stier et al. 2005) and as such should have a strong effect on its vertical profile. There are several uncertainties associated with the removal rates, from the raindrop-aerosol collision efficiency (Seinfeld and Pandis 2016) to the sub-grid co-variability between precipitation and aerosol (Gryspeerdt et al. 2015). In order to explore the effect of these uncertainties, we scale the in- and below-cloud removal tendencies up and down by a factor of 2. This large perturbation has the effect of changing the BC mean lifetime, for example, from a baseline value of ~7 days to ~5 and ~9 days for increased and decreased removal rates respectively. Figure 8 shows profiles of fractional bias for increased and decreased wet-deposition removal rates. The importance of wet-deposition in controlling the vertical distribution of the aerosol is immediately apparent. While increasing the wet-deposition rates leads to much stronger low biases in all cases, reducing the wet-deposition leads to a dramatic improvement. In the southern extratropics the size bias is virtually eliminated, although some smaller biases do remain. In the tropics the bias is also reduced, although in the boundary layer the larger aerosol is now *overestimated*. The large near-surface bias in the tropics remains unchanged, further suggesting that this bias is due to insufficient sources rather than over-efficient removal. In the northern extratropics the bias in larger particles is virtually eliminated in the free troposphere, at the expense of the smaller particles however, which now show a low bias, presumably due to the reductions in nucleation through additional condensational sink. This is suggestive that wet deposition is generally over-efficient in HAM, but that this is not the only source of the biases shown in Figure 4c, and that a simple global scaling of the removal rate would be unphysical and probably not be an effective solution. While biases in the precipitation rate could explain some of the aerosol biases, impaction scavenging is a relatively inefficient removal mechanism and ECHAM-HAM reproduces global patterns of precipitation reasonably well (e.g. Kipling et al. 2017). Biases in the in-cloud nucleation scavenging, which is a far more efficient mechanism, are therefor the most likely cause.

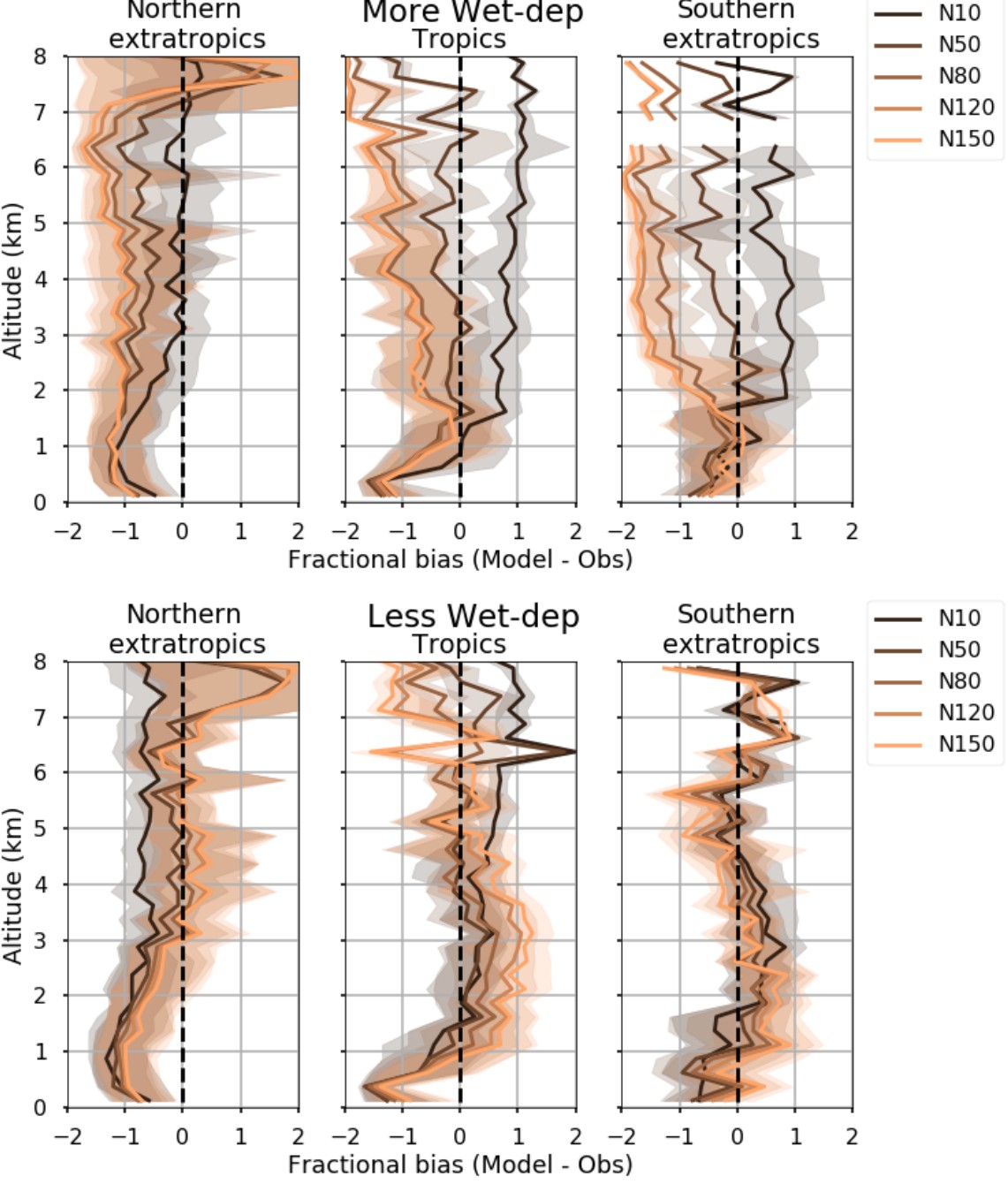

**Figure 8: The vertical profile of fractional bias in modelled aerosol number at different size cut-offs for measurements in the Northern extratropics (a), the Tropics (b), and the Southern extratropics (c) with increased (x2) and decreased (x0.5) wet-deposition rate.**

**Vertical flux in convection**

In order to determine the importance of convection on the vertical distribution of aerosol we scale the convective tracer entrainment by a factor of 10 up and down. This large perturbation causes a relatively small response in the vertical number size distribution, as shown in Figure 9. The largest effect is in the Tropics where reducing the tracer entrainment leads to a reduced bias in the free-troposphere for both small and larger particles. The reduced entrainment leads to a positive bias for larger particles in the boundary layer however. There are also small improvements in the extra-tropics. Increasing the tracer entrainment leads to increased biases throughout. The reduced entrainment leads to a lower concentration of N10 in the UT, corroborating previous work which showed that the MIT-CAM model had too-large a transport of CN into the upper-troposphere (Ekman et al. 2012).

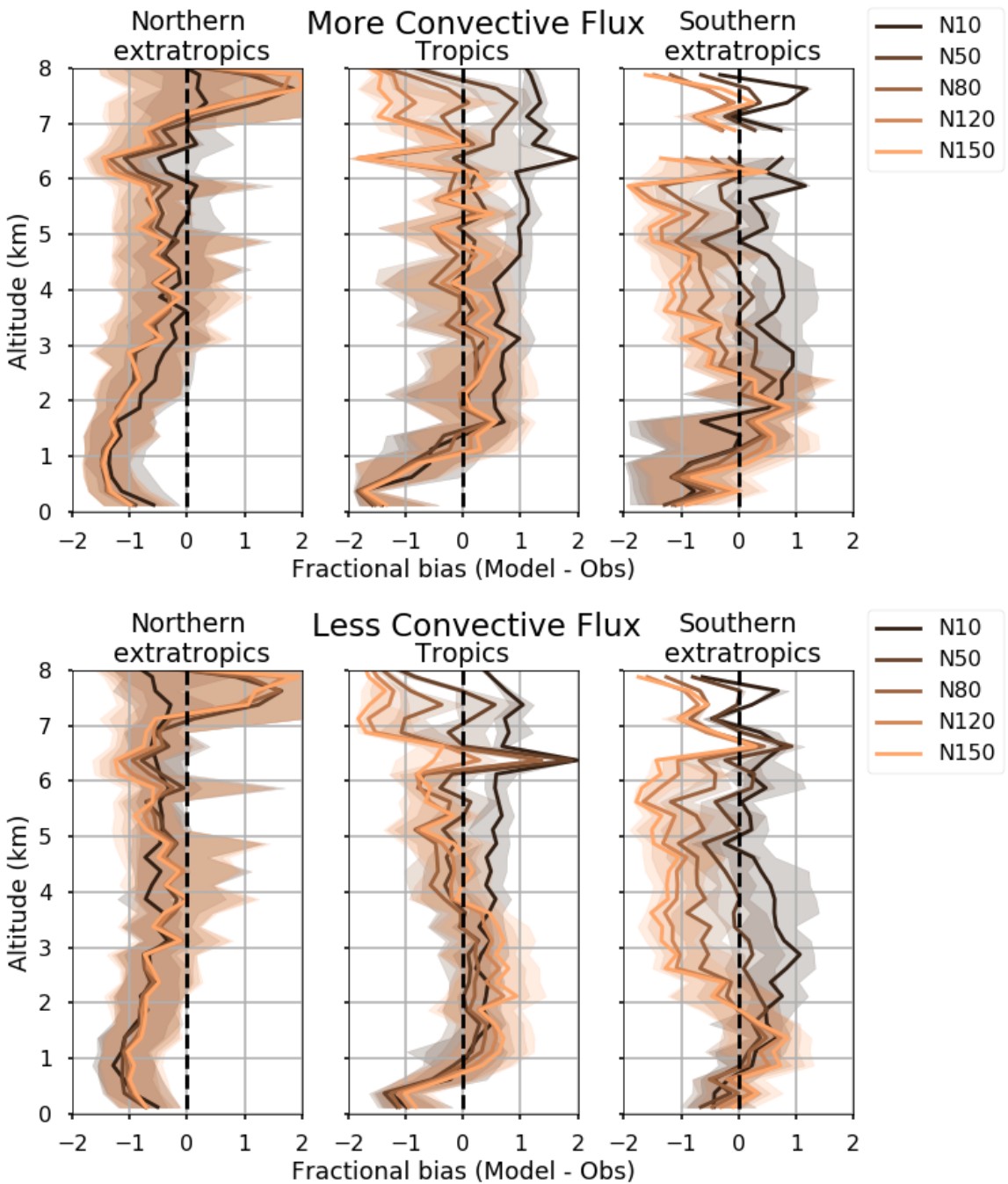

**Figure 9: The vertical profile of fractional bias in modelled aerosol number at different size cut-offs for measurements in the Northern extratropics (a), the Tropics (b), and the Southern extratropics (c) with increased (x10) and decreased (x0.1) convective tracer entrainment rate.**

**Coagulation**

Inter-mode coagulation provides another mechanism by which aerosol can be transferred from insoluble to soluble modes (by coagulating with a particle already in a soluble mode), and intra-mode coagulation provides a key growth pathway for Aitken and accumulation mode aerosol. Both of these will affect the vertical aerosol size distribution, and here we scale both intra-

5 and inter-mode coagulation by a factor of two. Figure 10 shows that increasing the coagulation rates reduces the low model bias throughout the troposphere in the tropics and southern extra-tropics for both larger and smaller particles. The largest improvement is in the tropical free troposphere where coagulation is a dominant mechanism for transfer of number from the nucleation to Aitken and Aitken to accumulation modes, due to the high number densities here (Schutgens and Stier 2014). Generally, decreasing the coagulation rates increases the model bias, apart from the boundary layer in the northern extra-

10 tropics where the decrease leads to a small reduction in the bias.

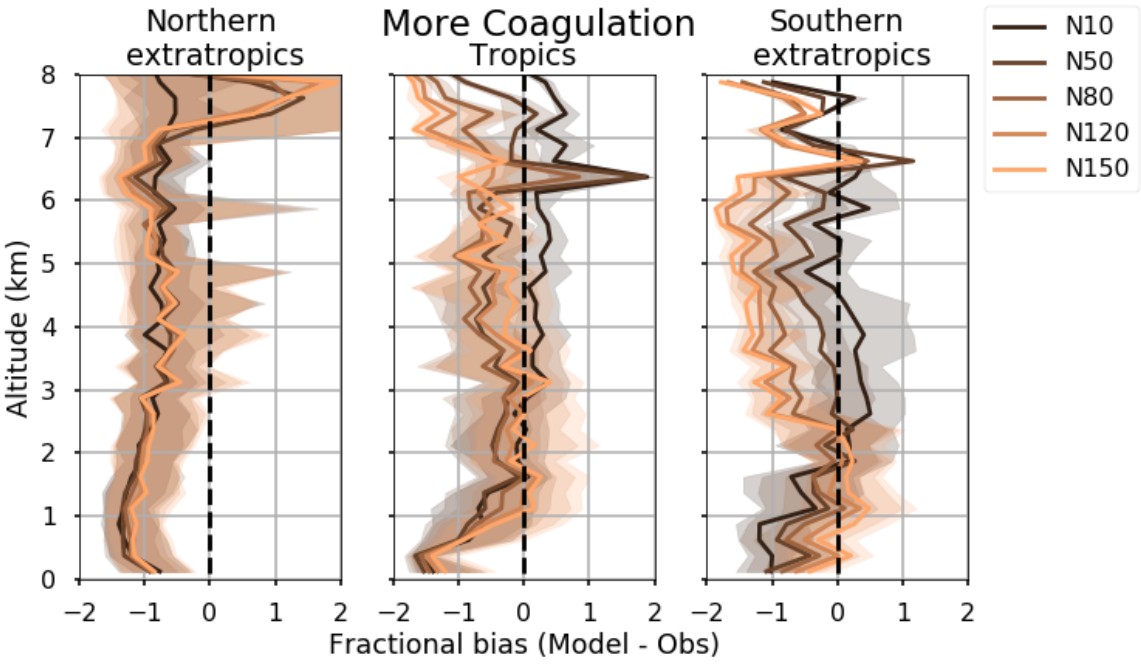

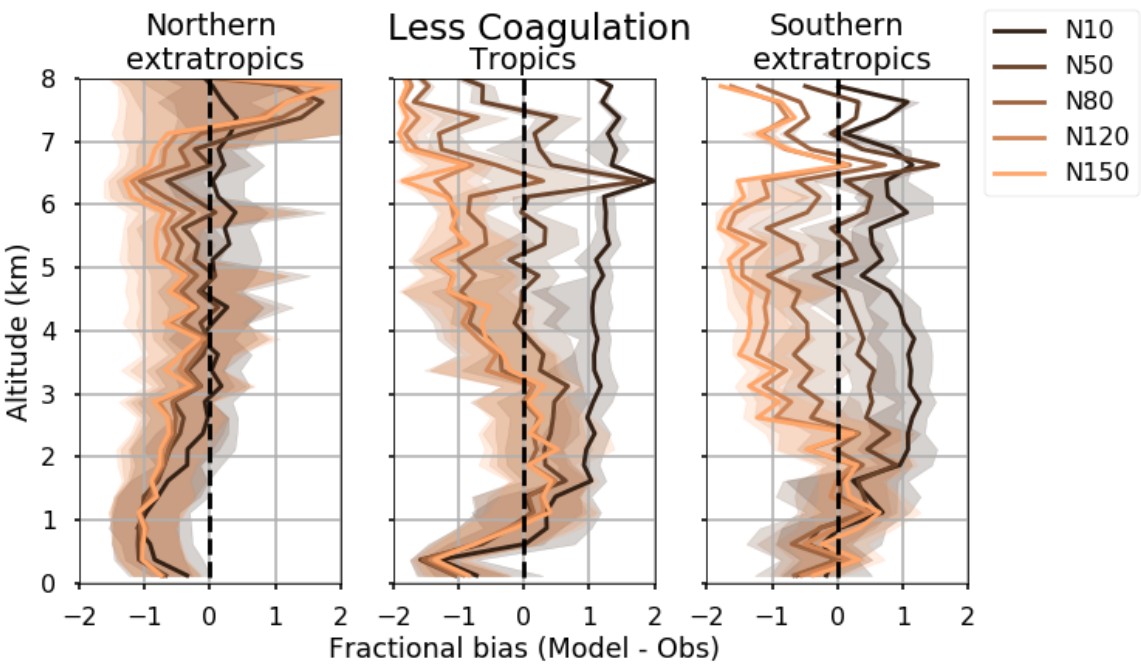

**Figure 10: The vertical profile of fractional bias in modelled aerosol number at different size cut-offs for measurements in the Northern extratropics (a), the Tropics (b), and the Southern extratropics (c) with increased (x2.0) and decreased (x0.5) coagulation rate.**

**Dry deposition**

Uncertainty in dry deposition has been shown to provide one of the largest contributions to uncertainty in the surface distribution of CCN in HadGEM-GLOMAP (Lee et al. 2013). However, despite scaling the dry-deposition rates for both the Aitken and accumulation modes over the same ranges, we see no significant change in the distribution of aerosol compared to the aircraft measurements (see figures in Appendix C). This could be due to the treatment of dry deposition as the lower boundary of the vertical diffusion scheme in ECHAM-HAM (Stier et al. 2005), which minimises spurious surface effects.

**4.3 Comparison with SALSA**

When aerosols grow due to various processes they can move between modes, e.g. from Aitken to accumulation. Another possible reason for the biases observed in the upper FT is that M7, the default aerosol scheme in ECHAM-HAM, has to perform a redistribution of number between modes, in order to avoid numerical diffusion, often referred to as 'mode merging'. This can result in 'stiff' modes which do not grow or shrink as efficiently as they should. However the SALSA bin scheme (H. Kokkola et al. 2008) is also available to use in ECHAM-HAM. Rather than representing the aerosol population as 7 log-normal modes as in M7, SALSA uses 20 bins in the standard configuration.

The model aerosol fields are interpolated onto the observational points as with M7 and the integrated number can be calculated directly by summing the appropriate aerosol bins. The median fractional bias in the integrated number as a function of altitude is shown in Figure 11. Interestingly, the SALSA aerosol scheme shows a similar negative bias in the large-particle concentration in the upper FT which suggests the mode merging in M7 is not the cause of the bias in ECHAM-HAM. SALSA also has a small positive bias in smaller particles not present in M7, which may be due to differences in the wet deposition scheme (T. Bergman et al. 2012). This result suggests that microphysical details can be of secondary importance compared to other physical processes, in particular wet deposition, when it comes to accurately representing the aerosol size distribution. A similar conclusion was reached when investigating the difference between bin and modal schemes in the GLOMAP model (Mann et al. 2012).

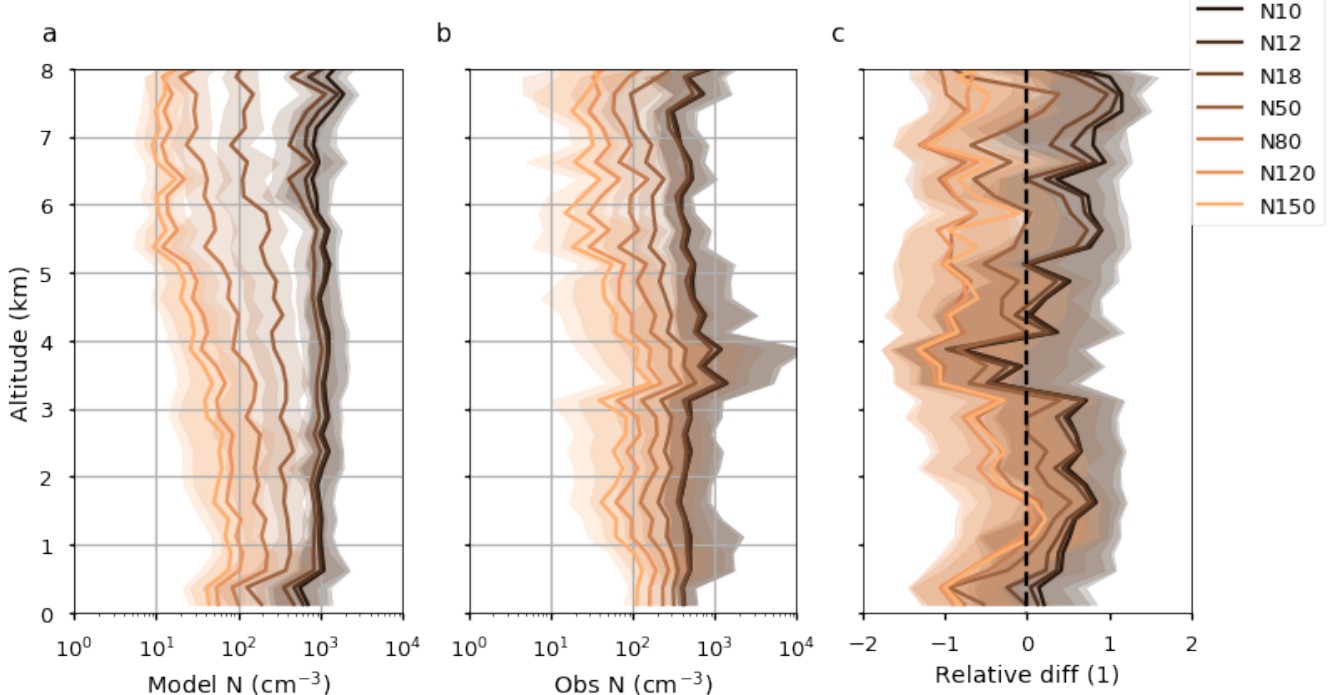

**Figure 11: The vertical profile of median fractional bias in model and observed integrated aerosol number at different size cut-offs using SALSA.**

**4.4 CCN**

Many of the aircraft included in the NSD analysis above also carried a CCN counter which is able to measure CCN either at specific or across a range of supersaturations (see Table 1 for details). By taking all of the measurements at each supersaturation and comparing with the model CCN at the same supersaturation we are able to create profiles of the fractional bias in CCN at a range of frequently measured supersaturations, as shown in Figure 12.

The CCN profiles contain fewer measurements since not all the flights carried a CCNC and some of these instruments 'scanned'
across supersaturations and hence only measured at any given supersaturation a fraction of the time. As discussed in the introduction, the CCN spectra also depend on both the aerosol size distribution and hygroscopicity, but it can be clearly seen that the bias in the NSD shown in Section 4.1 manifests itself in a low bias in the CCN at lower supersaturation (mostly larger particles) in the FT. The same processes identified through the sensitivity analysis as being important for influencing the vertical size distribution, namely wet deposition and condensational growth, control the vertical CCN spectra. Although many
cloud regimes are updraft limited rather than CCN limited (Reutter et al. 2009) this low bias in low-supersaturation CCN is likely to have an important impact on the forcing in those CCN-limited regimes.

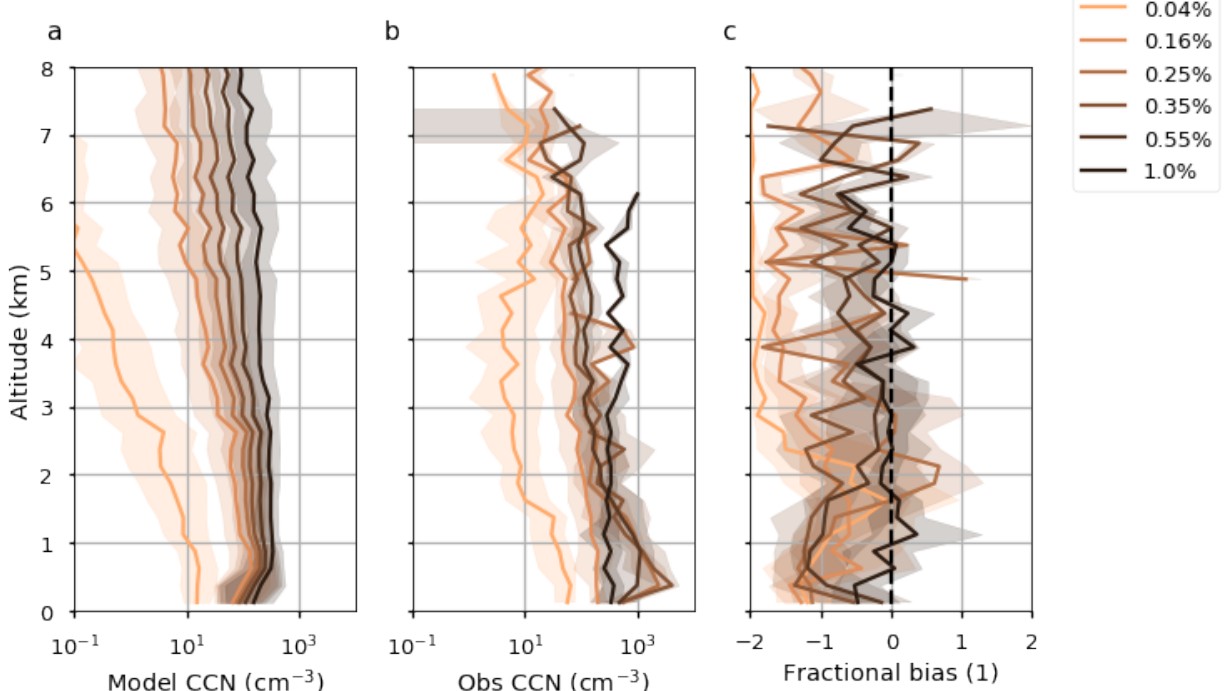

**Figure 12: The vertical profile of median collocated model (a), observed (b) and fractional bias (c) in CCN at the measured supersaturation, as shown in the legend.**

## 5 Discussion and conclusions

We have evaluated the vertical size distribution of sub-micron aerosol particles in the global aerosol model ECHAM-HAM using a dataset of in-situ aircraft measurements that covers large parts of the globe. The model generally performs well considering the challenges in reproducing in-situ observations with a global model but shows a negative bias in accumulation mode aerosol in the mid- to upper-troposphere, although due to the exponential decrease in pressure the absolute biases in concentration are still small. By comparing the bias over land and ocean we show that this bias could result from errors in the ageing and removal processes rather than in the emissions. This bias in particle concentrations translates into a negative bias for low-supersaturation CCN at similar altitudes. The model also underestimates marine sulfate in the boundary layer, likely due to an under-representation of DMS emissions. A similar bias, which contributed to an overly large aerosol forcing, has been seen in UKESM1 (Mulcahy et al. 2018) and will be explored in other models in future work.

We also performed a simple one-at-a-time parameter perturbation study (summarised in Appendix B Figure 14) which showed that wet deposition, a key aerosol removal process in ECHAM-HAM, is probably over-efficient, particularly in the southern extra-tropics. One potential reason for this over-efficiency is the assumption (common in GCMs) that aerosol mixes instantaneously across a grid box (Gryspeerdt et al. 2014). Both tracer entrainment and coagulation are shown to be important mechanisms controlling the vertical distribution of aerosol in the tropics, but with limited impact elsewhere. The modelled

aerosol size distribution shows reduced bias compared to the aircraft measurements when these processes are tuned up and down respectively. Condensational growth is particularly important in the northern extra-tropics where slower ageing (requiring an increased amount of sulfate) reduces the model bias. A similar sensitivity analysis for HadGEM-UKCA (Kipling et al. 2016) showed qualitatively similar results: reduced condensational growth led to fewer small particles and more large

particles in the upper troposphere; and coagulation having the greatest effect on particle concentrations in the tropics. In this study, however, ECHAM-HAM does not show the pronounced effect of convective entrainment or dry deposition seen in HadGEM-UKCA. This could be due to the different convective parameterisations used in each model and the treatment of dry deposition as a lower boundary of the vertical diffusion scheme in ECHAM-HAM, which minimises spurious surface effects. These simple perturbations do not allow us to explore the complex interactions between these processes, but they do

demonstrate the magnitude of the single effects, and they highlight the value of these measurements in evaluating them. By performing a full sampling of these parameterisations and combining the constraints developed in this work with other remote-sensing datasets it will be possible to significantly improve our confidence in the representation of aerosol in ECHAM-HAM. The increasing availability of aircraft datasets measuring the vertical distribution of aerosol, particularly in the UT, provides valuable constraints for GCMs, with implications for improving our representation of aerosol direct and indirect effects in

these models.

The single model year used in the present evaluation introduces some uncertainty in the representativeness of the model values when comparing with in-situ measurements. The sensitivity analyses performed also depend on the particular representation of specific processes used in ECHAM-HAM and may behave differently in other GCMs (structural uncertainty). A multi-model multi-year experiment to apply these constraints within the AeroCom framework and explore inter-model biases and

structural uncertainties is currently underway. This will also include measurements from the ATOm (Atmospheric Tomography) campaign, designed specifically to explore the vertical distribution of aerosol and pre-cursor gasses in the remote atmosphere (Wofsy et al., 2018). Further, by combining the number size distribution data used here with speciated mass concentration data from e.g. AMS measurements it should be possible to provide detailed insight into any deficiencies in the aerosol lifecycle in these models.

**Acknowledgements**

The authors thank Jens Redemann, Jamie Trenbeth, Harri Kokkola and Dan Partridge for useful discussions.

DWP and PS acknowledge funding from Natural Environment Research Council projects NE/J022624/1 (GASSP), NE/L01355X/1 (CLARIFY), NE/M017206/1 (IMPALA) and NE/P013406/1 (A-CURE) and from the Science and Technology Facilities Council project ST/P003206/1 (EVADE). PS also acknowledges funding from the European Research Council

project RECAP under the European Union's Horizon 2020 research and innovation programme with grant agreement 724602 and the European Union's Seventh Framework Programme (FP7/2007-2013) projects BACCHUS under grant agreement 603445. KSC acknowledges funding from the Natural Environment Research Council projects NE/J022624/1 (GASSP),

NE/L01355X/1 (CLARIFY) and NE/P013406/1 (A-CURE). The ECHAM-HAMMOZ model is developed by a consortium composed of ETH Zurich, Max Planck Institut für Meteorologie, Forschungszentrum Jülich, University of Oxford, the Finnish Meteorological Institute and the Leibniz Institute for Tropospheric Research, and managed by the Center for Climate Systems Modeling (C2SM) at ETH Zurich. The ECHAM-HAM simulations were performed using the ARCHER UK National

5 Supercomputing Service. FAAM airborne data were obtained using the BAe-146 Atmospheric Research Aircraft, which was operated by Airtask and jointly funded by the UK Natural Environment Research Council (NERC) and the Met Office. We are grateful to the many PIs who contributed their data to GASSP: T. Choularton (ACCACIA, COPE); J. Hudson (ACE1, INDOEX); A. Clarke (ACE1, ACEASIA / ACE-Asia, ARCTAS, INDOEX, INTEX-A, MIRAGE, PASE, PEM Tropics A+B, VOCALS-REx); H. Coe (AMMA, AEGEAN-GAME, BORTAS, EUCAARI-LONGREX, OP3, RONOCO, VOCALS-REx);

10 J. Trembath (APPRAISE, EM25, EUCAARI-LONGREX, RONOCO, TROMPEX, VOCALS-REx); A. Nenes (ARCPAC, ARCTAS, CALNEX, DC3, DISCOVER-AQ, SEAC4RS, TEXAQS-GoMACCS 2006); A. Middlebrook (ARCPAC, CALNEX, TEXAQS-GoMACCS 2006); R. Bahreini (ARCPAC, CALNEX, TEXAQS-GoMACCS 2006); J. Jimenez (ARCTAS, DC3, INTEX-B, MIRAGE, SEAC4RS); B. E. Anderson (DISCOVER-AQ, INTEX-A); J. M. Comstock (GOAmazon); F. Mei      (GOAmazon); R. Weber (INTEX-A); G. Roberts (INTEX-B, MIRAGE); S. Howell (PASE,

15 VOCALS-REx); D. J. Jacob (PEM Tropics A); G. M. Gardner (PEM Tropics A); M. G. Schultz (PEM Tropics A); R. Talbot (PEM Tropics A, PEM West A); and J. Snider (VOCALS-REx).

The authors declare that they have no conflict of interest.

**Author contributions**

DWP and PS designed the experiments and DWP carried them out. DWP and NS performed the simulations. CR, KP, DL, JA,

20 HC and KC provided the formatted and standardised aircraft data. DWP prepared the manuscript with contributions from all co-authors.

**Appendix A**

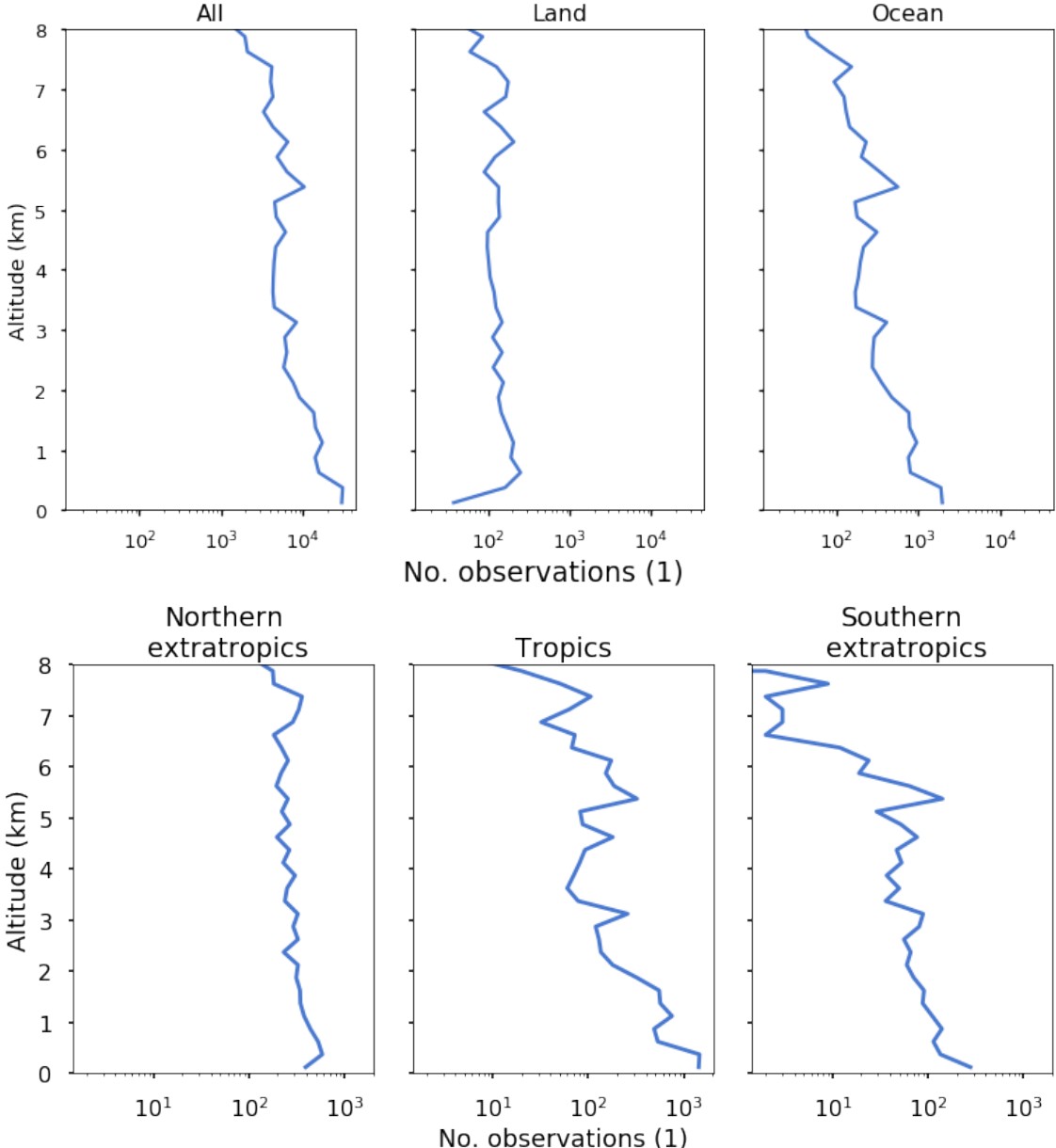

5   **Figure 13: The number of observations (and model points) used in each altitude bin for different subsets of the data used throughout the analysis.**

**Appendix B**

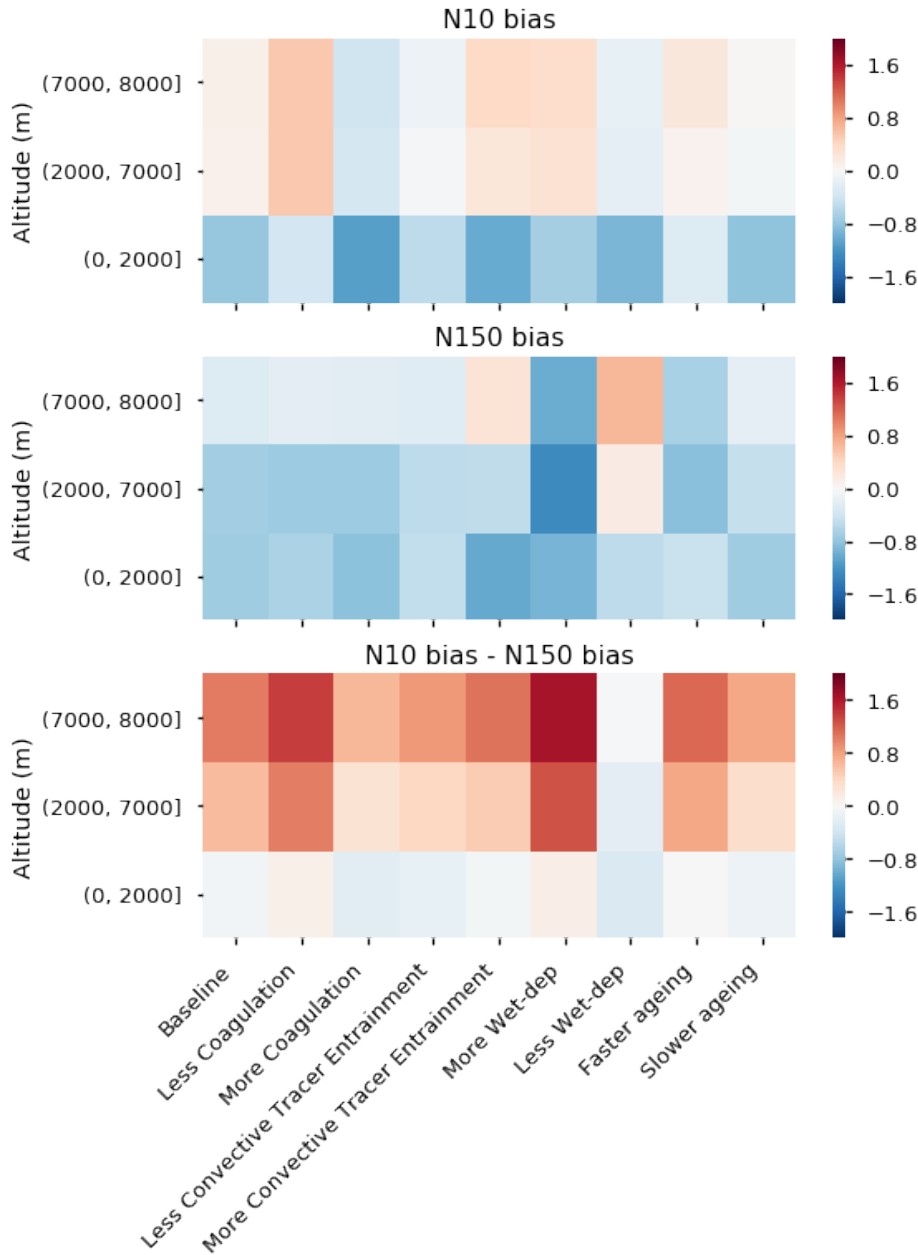

**Figure 14: Summaries of the fractional bias (Model – Obs) in N10 and N150 for each of the sensitivity experiments across three (unequal) altitude ranges. The difference between bias in N10 and N150 is also shown.**

**Appendix C**

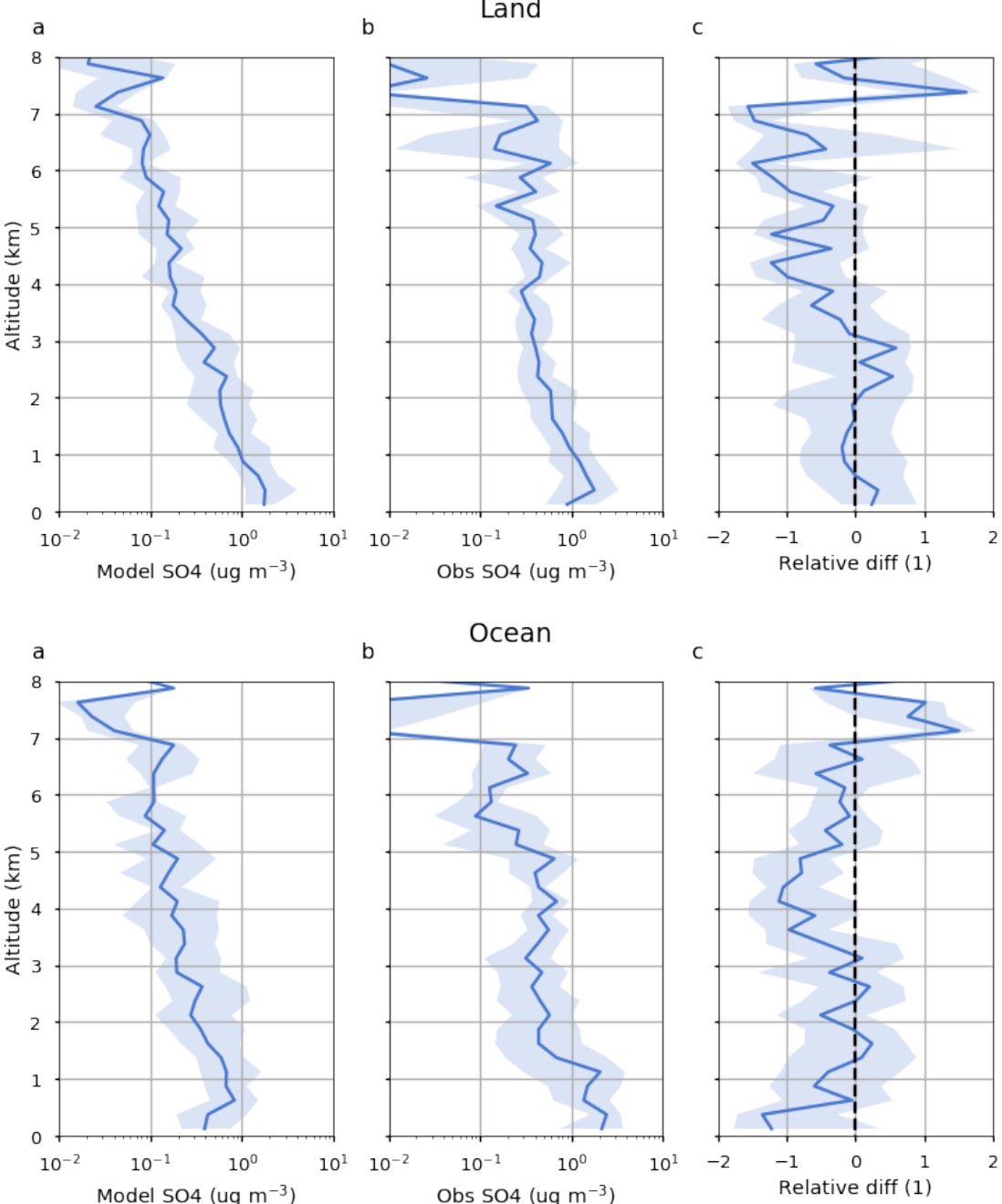

**Figure 15: The vertical profiles of modelled (a), observed (b) and fractional bias (c) in SO4 mass concentration for measurements over Land and Ocean.**

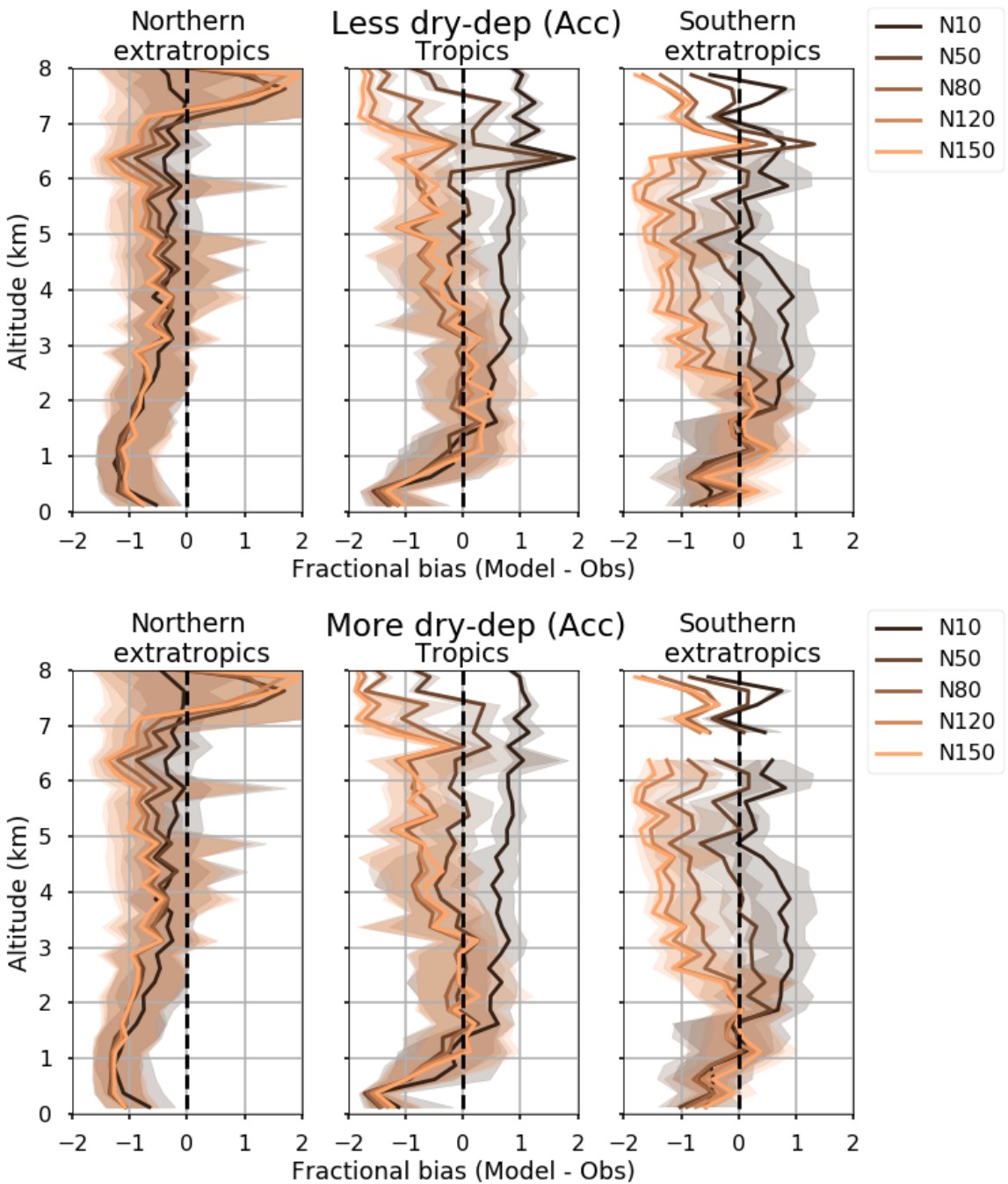

**Figure 16: The vertical profile of fractional bias in modelled aerosol number at different size cut-offs for measurements in the Northern extratropics (a), the Tropics (b), and the Southern extratropics (c) with reduced (x0.1) and increased (x10) dry-deposition in the accumulation mode.**

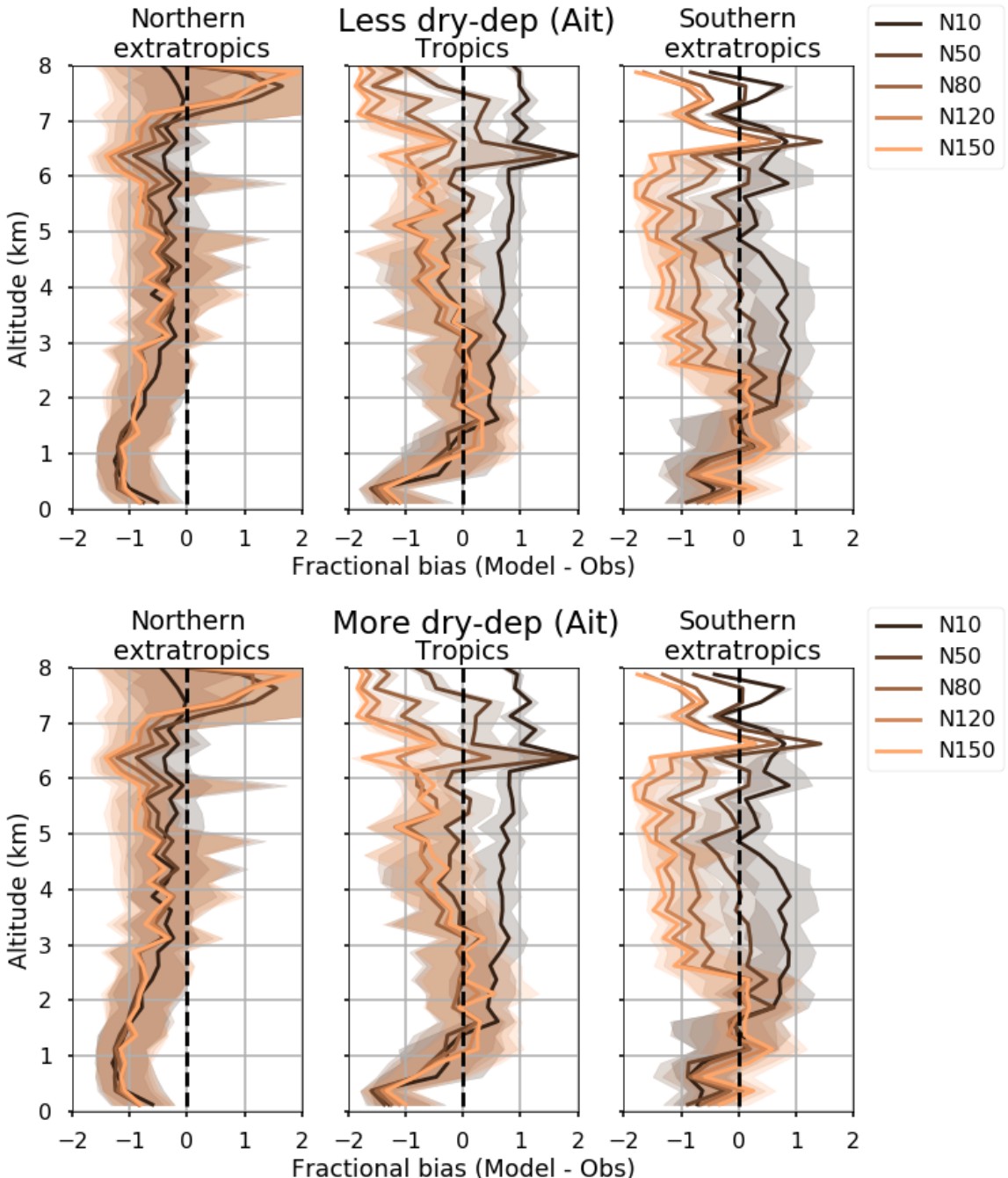

**Figure 17: The vertical profile of fractional bias in modelled aerosol number at different size cut-offs for measurements in the Northern extratropics (a), the Tropics (b), and the Southern extratropics (c) with reduced (x0.5) and increased (x2.0) dry-deposition in the Aitken mod.**

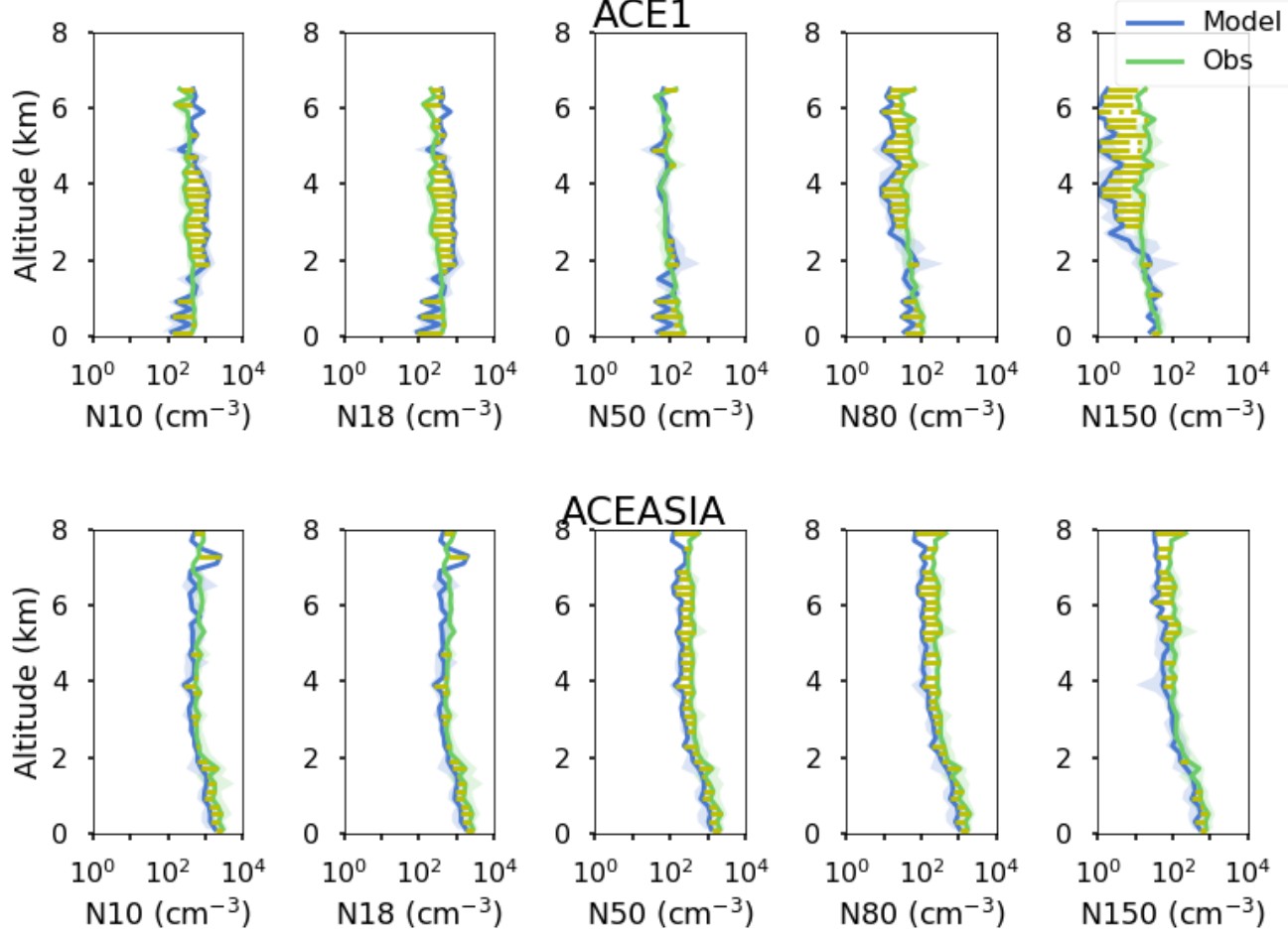

**Figure 18: Per-campaign median profiles of aerosol number at different sizes. Yellow lines indicate a significant difference at that altitude.**

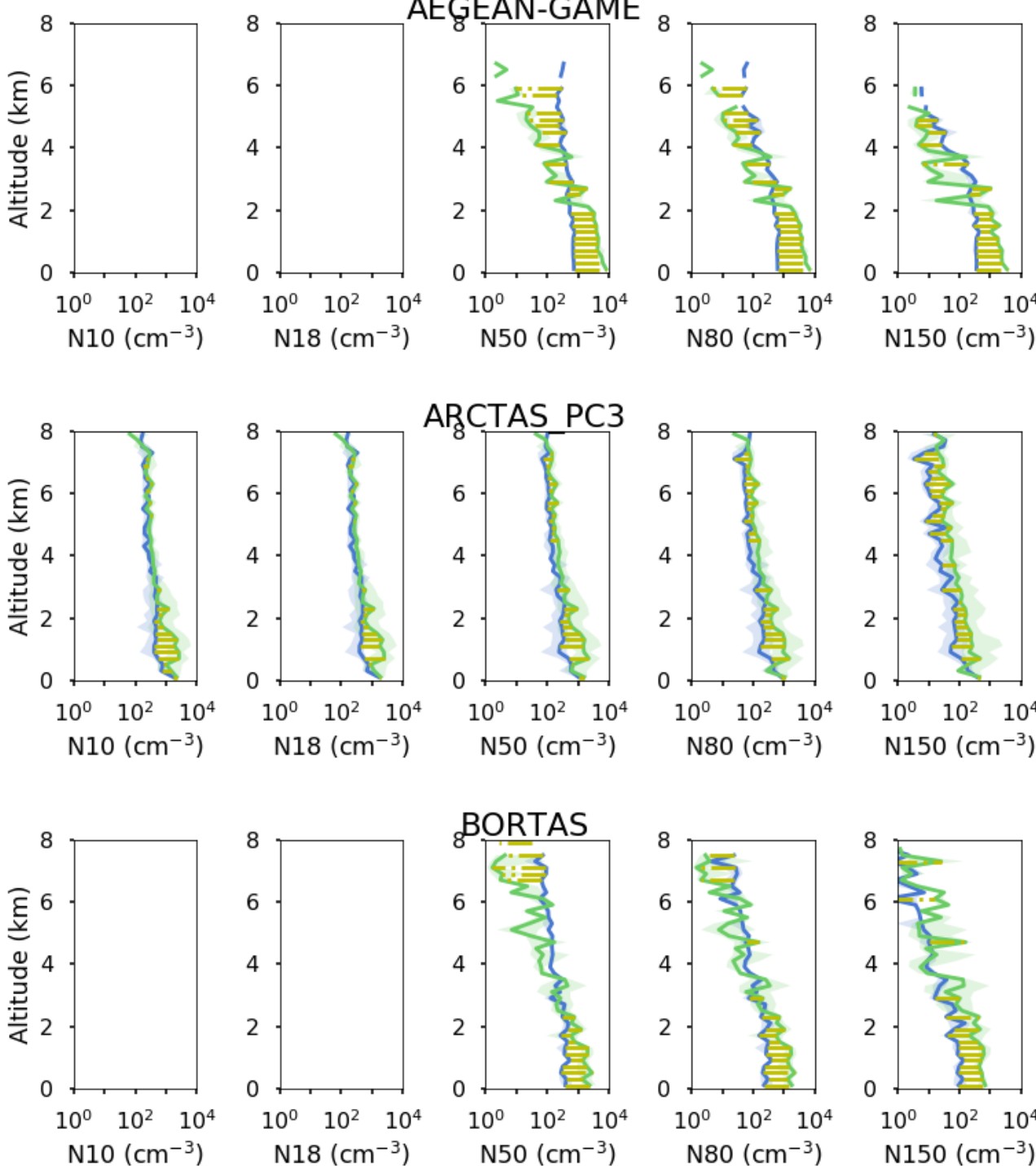

**Figure 18 cont.**

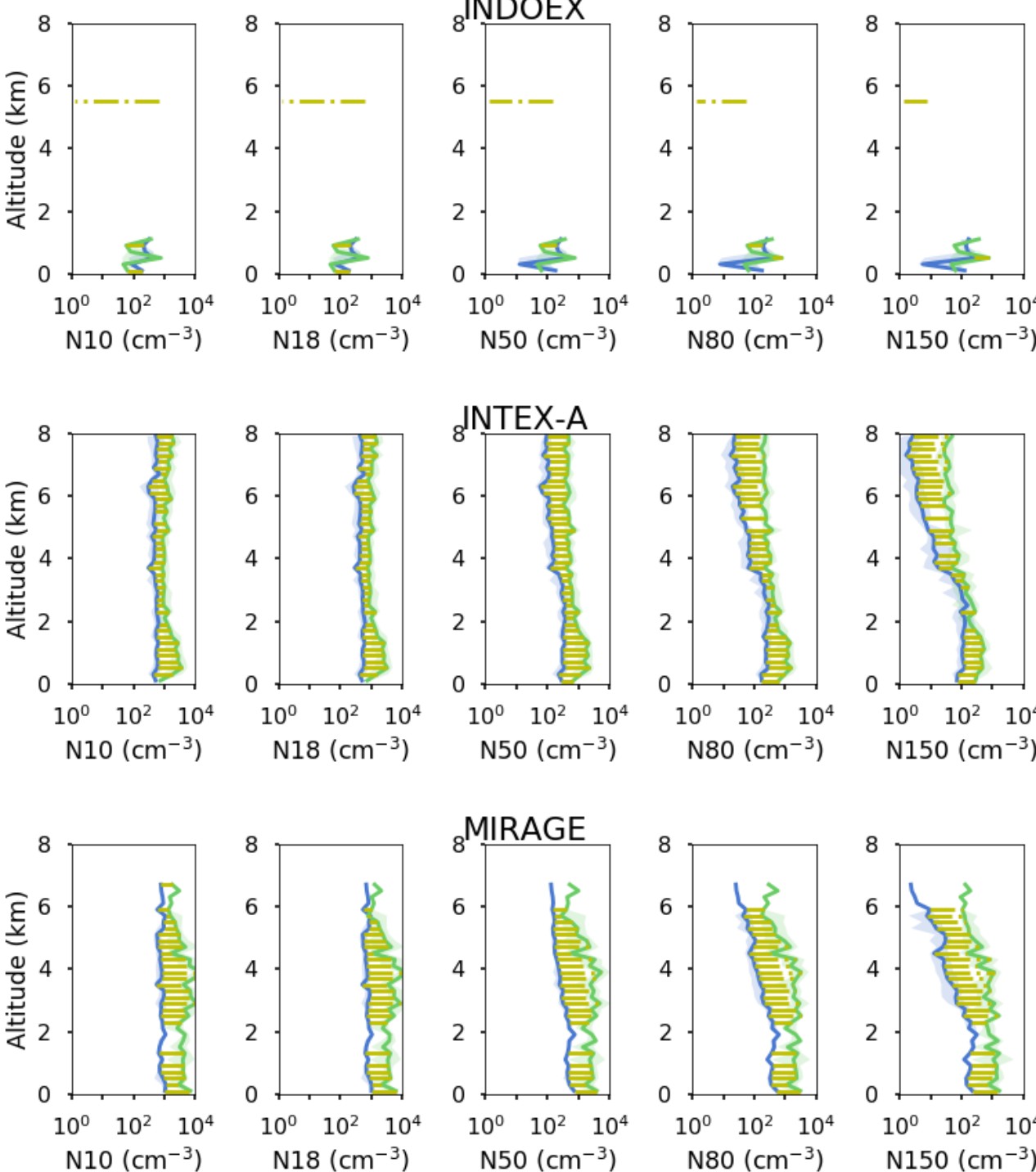

**Figure 18 cont.**

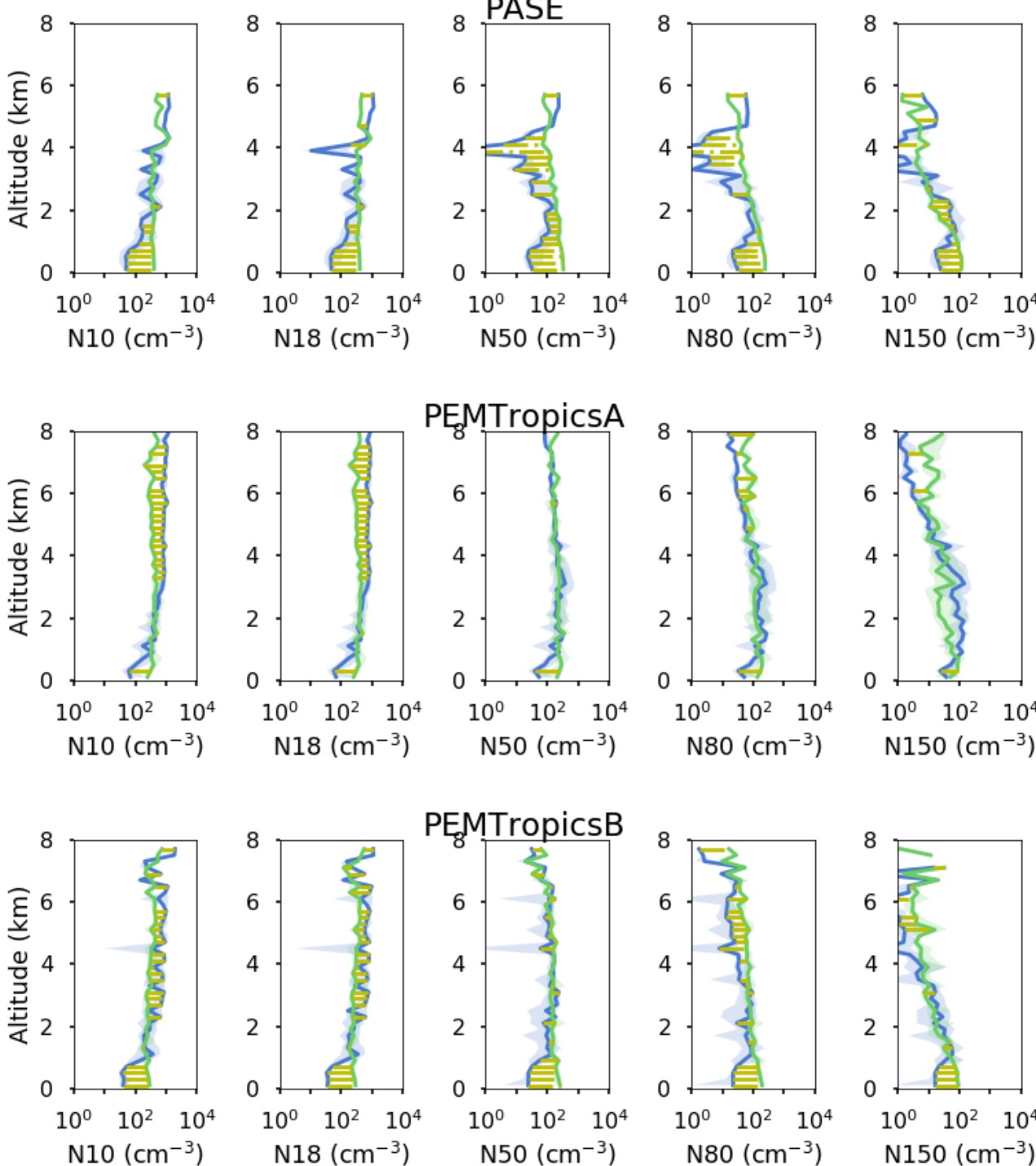

**Figure 18 cont.**

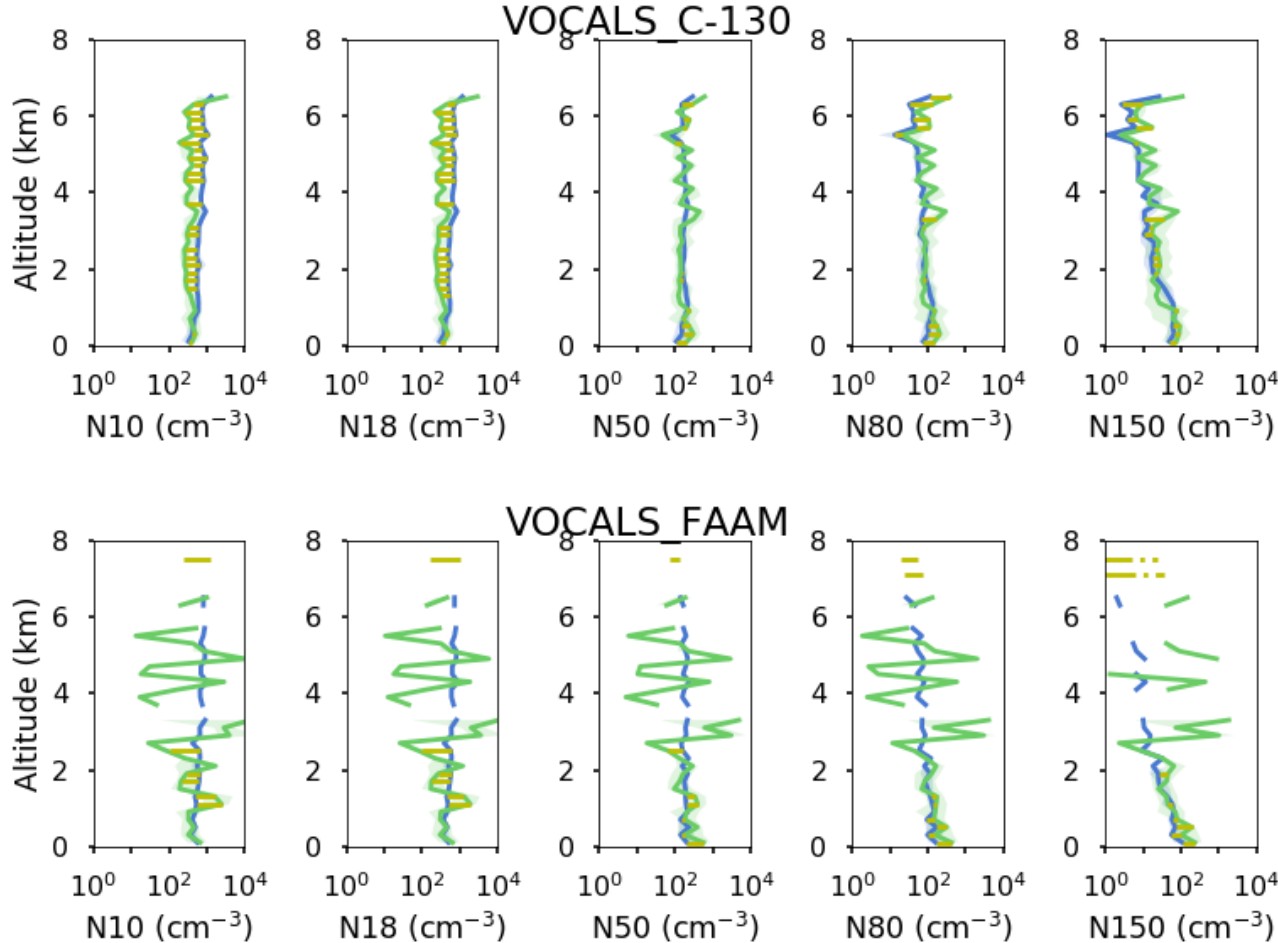

**Figure 18 cont.**

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
