# Peer review of "In-situ constraints on the vertical distribution of global aerosol"

_Atmospheric Chemistry and Physics, 2018_

## Referee Comment (RC1) · Anonymous Referee #1 · 1 Mar 2019

The vertical distribution of aerosol is a critical feature to know for enhancing our understanding of aerosol life cycle, estimating climate forcing of aerosols with confidence, and constraining numerical models to better reproduce past and project future climate. Nevertheless, it is still poorly known primarily due to the lack of observations. The authors have compiled a database, the Global Aerosol Synthesis and Science Project (GASSP) from past airborne observations. They have also overcome certain statistical issues in making the data useful for constraining models in an adequate procedure. This paper presents a case of applying GASSP to evaluate the performance of the global climate model ECHAM-HAM that includes a modal aerosol model (M7) as well as a sectional aerosol model (SALSA).

The model simulations were designed straightforwardly, including a set of single-

parameter-perturbation simulations in addition to a standard case run for exploring the sensitivity of the modeled results to certain physical or chemical parameters in a rather simplistic way. The comparison between modeled and observed data covers vertical profiles of aerosol and CCN size distributions over several selected regions, serving a good purpose for identifying the model biases while as a demonstration of using GASSP to constrain models. The paper is well organized, and the result is largely presented clearly. The content of the paper is definitely suitable for the readers of ACP. The work is also informative to the similar efforts in near future. Nevertheless, there still are some issues need to be adequately addressed before the acceptance of the paper for publication.

The authors analyzed certain reasons that could cause the biases of the model, including wet removal of aerosols by precipitating particles and removal through nucleation scavenging. Since the comparison is against observations, this discussion hence should not just be limited to the processes included in the model, but also potentially important ones presently excluded in the model. Regarding the model bias in underestimating free tropospheric aerosol number concentration, there is another factor, i.e., resuspension of aerosols resulting from evaporation of cloud drops. This could be, as indicated by previous works (e.g., Hoppel et.al, 1994 JGR, D7 14443 and beyond for measurement; Grandey et al., 2018, ACP, 15783 and Kim et al., 2008, JGR D16309 for global modeling), an important source for accumulation mode aerosols in the free troposphere. Therefore, such an effect should be discussed. If this process is not included in ECHAM-HAM (perhaps the majority of global models do not include it anyway), the authors need at least mentioning the limitation of their analyses due to this reason.

The statement in Page 8 line 12 that "the inter-annual variability in aerosol burden ,. . . is small" appears to be made without considering the common feature of inter-annual variation of precipitation.

In comparing modeled with observational data, the authors need to provide several

additional details: (a) the number of observational samples for each of the selected regions (e.g., Fig. 3); (b) objective scores of fractional bias, e.g., vertically accumulated absolute bias in order to make a better judgement on the overall model performance in comparison with other cases.

A few specific comments.

In page 14, line 9, "20nm" should be "10 nm"?

In page 17, line 12-13, the sentence of "but by requiring . . . ageing profiles" is difficult to understand.

In page 19, line 8-9, "reducing the wet-deposition . . .", could the authors provide an estimate of a corresponding change in aerosol lifetime?

In page 28, line 4-7, while the authors suggest that "ECHAM-HAM does not show . . . convective entrainment or dry deposition. . .", the following sentence only provides a discussion related to dry deposition.

Figure quality: in many figures, the legend often overlaps with plotting area with data.

---

## Referee Comment (RC2) · Anonymous Referee #2 · 7 Mar 2019

This manuscript uses the GASSP database of airborne measurements to evaluate the aerosol number distribution simulated with the ECHAM-HAM model around the world. It then compares a series of sensitivity experiments which test physical processes controlling the loss of aerosol against these observations. The manuscript is generally clear and the topic is certainly relevant to ACP. In general, the paper needs more detail on necessary background and to substantiate conclusions. Specific comments follow:

1. The model description is incomplete. The manuscript should describe exactly which aerosol sources are simulated (e.g. OA, BC, nitrate, etc.) and previous efforts to evaluate these schemes. For example, the discussion of the ECHAM treatment of SOA on page 28 should have been included in the model description.

2. The sensitivity tests focus on only a subset of physical processes, largely limited

to loss processes. The authors should acknowledge the role that many other processes play in dictating aerosol number concentrations, including oxidation rates, nucleation, thermodynamic partitioning, and obviously, emissions. It seems equally likely that any/all of these processes could contribute to model bias. The importance of these various processes in the bias could be untangled with an evaluation of speciated mass concentrations. The authors might highlight these complementary approaches (i.e. that number concentration evaluation is perhaps the most relevant metric for climate, but that speciated mass concentrations can provide greater insight into aerosol sources/formation) in both the Introduction (in the paragraph page 2, lines 17-29) and in the Conclusions as a next step to evaluating model fidelity. I note that sulfate mass concentration comparisons are shown in the Appendix – why did the authors not include evaluation of the other species measured by AMS?

3. Page 8, line 11: The statement that "interannual variability in aerosol burden is small" may be true, but the long-term trend is not and given that measurements from GASSP extend almost 2 decades, this statement gives a false impression that variations in emissions are not relevant to these comparisons. This is an inherent weakness of this study (comparisons with only one model year), and in absence of a more detailed analysis of how meteorological and emissions variations over two decades contributes to model-measurement airborne point comparisons, the authors must acknowledge that their comparison is not "apples-to-apples".

4. Similarly, Section 3 explores the temporal sampling aspect of model evaluation, but does not address the fundamental temporal mis-match. It may be more appropriate to evaluate a free-running GCM using a 10 year simulation to capture the role of interannual variation in meteorology and/or emissions, particularly on comparisons with temporally-limited, localized campaigns. Could the authors comment on this?

5. Section 3: Could the authors also comment on why they did not simply average the GASSP measurement to the model spatial resolution (in order to not penalize the model for its inability to reproduce sub-grid variability) rather than using 2 minute averages?

MINOR COMMENTS

1. Section 2.1: It would be helpful if the authors clearly describe how they will use the measurements in Table 1. It appears that they largely focus on the DMA, OPC, and SMPS measurements, with sulfate mass concentrations from the AMS shown in the Appendix.

2. Figure 1: flight tracks are illegible. Make flight tracks finer so that they don't appear as blotches, and consider different colour scheme that enables differentiation of campaigns in the same region.

3. Page 14, line 6: suggest replace "well" with "best" – biases that exceed 50% are not suggestive of a very good simulation. Similarly, the authors should temper their language on page 27, line 6.

4. Page 14, line 9: should this be 10nm? There is no 20 nm cut-off presented in Figure 4

5. Page 15, line 18: "near the surface" – suggest replace with "in the boundary layer" as these biases extend through several kms

6. Page 15, line 18: "local emissions sources not resolved by the coarse model resolution" seems an unlikely explanation for discrepancies. Sub-grid plumes may not be resolved by the model, but these should not impact medians. Furthermore, emissions inventories generally do include point source emissions and while specific localized sources may be missing from inventories, they are not de facto globally biased low.

7. Page 15, line 25: might insufficient marine organics also contribute to this bias?

8. Figures 5, 6: it would be useful to include the number of points in these sub-set comparisons on the figures.

---

## Referee Comment (RC3) · Anonymous Referee #3 · 13 Mar 2019

In-situ constraints on the vertical distribution of global aerosol by Watson-Parris et al. evaluates the size-and altitude-resolved aerosol number concentration produced by the ECHAM-HAM global model. The evaluation uses as reference dozens of airborne experiments carried out in the past two decades.

While modeled vertical profiles have been studied in the past, the large volume of observation and the focus on aerosol number size distribution make this study unique. The figures are clear. There are a couple of statements that need more explanations. I recommend publication after the authors consider the following items.

Page 2, line 2. The indirect forcing also depends on ice nuclei.

Page 2, line 12. Suggest also citing Kacenelenbogen, M., M. A. Vaughan, J. Redemann, R. M. Hoff, R. R. Rogers, R. A. Ferrare, P. B. Russell, C. A. Hostetler, J. W. Hair, and B. N. Holben. 2011. "An Accuracy Assessment of the CALIOP/CALIPSO Version 2/version 3 Daytime Aerosol Extinction Product Based on a Detailed Multi-Sensor, Multi-Platform Case Study." Atmospheric Chemistry and Physics 11 (8): 3981–4000.

Page 2, line 34. Insert "in situ" between aircraft and measurements.

Page 9. Line 25. omponents should read components.

Page 10., line19. VOCALS also did repeated sampling. Cite Wood, R., C. R. Mechoso, C. S. Bretherton, R. A. Weller, B. Huebert, F. Straneo, B. A. Albrecht, et al. 2011. "The VAMOS Ocean-Cloud-Atmosphere-Land Study Regional Experiment (VOCALS-REx): Goals, Platforms, and Field Operations." Atmospheric Chemistry and Physics 11 (2): 627–54.

Page 13, line 1. Clarke and Kapustin (2002) discuss the high number concentrations at >8 km altitudes in the tropics. Refer to it.

Page 15, line 11. Explain why the uncertainty in DMA charging results in erroneous counting but not sizing.

Page 17, line 14. "reduced negative bias" is taken to mean an increased number from the model. Explain why the slower condensational ageing shows this for larger particles.

---

## Author Comment (AC1) · 21 Jun 2019

**Response to reviewers**

We thank the reviewers for their critical feedback and insightful comments. We have responded to each of the comments in-line below and, where appropriate, indicated the relevant changes in the manuscript.

**Reviewer 1**

The authors analyzed certain reasons that could cause the biases of the model, including wet removal of aerosols by precipitating particles and removal through nucleation scavenging. Since the comparison is against observations, this discussion hence should not just be limited to the processes included in the model, but also potentially important ones presently excluded in the model. Regarding the model bias in underestimating free tropospheric aerosol number concentration, there is another factor, i.e., resuspension of aerosols resulting from evaporation of cloud drops. This could be, as indicated by previous works (e.g., Hoppel et.al, 1994 JGR, D7 14443 and beyond for measurement; Grandey et al., 2018, ACP, 15783 and Kim et al., 2008, JGR D16309 for global modeling), an important source for accumulation mode aerosols in the free troposphere. Therefore, such an effect should be discussed. If this process is not included in ECHAM-HAM (perhaps the majority of global models do not include it anyway), the authors need at least mentioning the limitation of their analyses due to this reason.

> *Our model does include the resuspension of aerosol resulting from the evaporation of precipitation – this has now been clarified in the model description. There are possible extensions of this simple treatment though which we now refer to both in the sensitivity analysis section (P9L27-P10L4), and the discussions where we explicitly discuss the focus of future work being the understanding of these structural model errors (P29L23-29).*

The statement in Page 8 line 12 that "the inter-annual variability in aerosol burden ,. . . is small" appears to be made without considering the common feature of inter-annual variation of precipitation.

> *This raises a good point, we've added an explicit acknowledgment of this uncertainty (P12L3-8) and clarified that this will be another focus of future work (P29L23-29).*

In comparing modeled with observational data, the authors need to provide several additional details: (a) the number of observational samples for each of the selected regions (e.g., Fig. 3); (b) objective scores of fractional bias, e.g., vertically accumulated absolute bias in order to make a better judgement on the overall model performance in comparison with other cases.

> *These have been added as Appendix A and B respectively.*

A few specific comments.
- In page 14, line 9, "20nm" should be "10 nm"?
- In page 17, line 12-13, the sentence of "but by requiring . . . ageing profiles" is difficult to understand.

- In page 19, line 8-9, "reducing the wet-deposition . . .", could the authors provide an estimate of a corresponding change in aerosol lifetime?
- In page 28, line 4-7, while the authors suggest that "ECHAM-HAM does not show . . . convective entrainment or dry deposition...", the following sentence only provides a discussion related to dry deposition.
- Figure quality: in many figures, the legend often overlaps with plotting area with data.

*Thank you for noting these, they have each been rectified in the latest version of the manuscript.*

**Reviewer 2**

1. The model description is incomplete. The manuscript should describe exactly which aerosol sources are simulated (e.g. OA, BC, nitrate, etc.) and previous efforts to evaluate these schemes. For example, the discussion of the ECHAM treatment of SOA on page 28 should have been included in the model description.

> *Yes, we agree, the model description was too brief – this has been extended and the effect of the prescribed SOAs discussed in more detail in Section 2.2.*

2. The sensitivity tests focus on only a subset of physical processes, largely limited to loss processes. The authors should acknowledge the role that many other processes play in dictating aerosol number concentrations, including oxidation rates, nucleation, thermodynamic partitioning, and obviously, emissions. It seems equally likely that any/all of these processes could contribute to model bias. The importance of these various processes in the bias could be untangled with an evaluation of speciated mass concentrations. The authors might highlight these complementary approaches (i.e. that number concentration evaluation is perhaps the most relevant metric for climate, but that speciated mass concentrations can provide greater insight into aerosol sources/formation) in both the Introduction (in the paragraph page 2, lines 17-29) and in the Conclusions as a next step to evaluating model fidelity. I note that sulfate mass concentration comparisons are shown in the Appendix – why did the authors not include evaluation of the other species measured by AMS?

> *Based on this and similar feedback from Reviewer 1, we have added a more explicit discussion of the potential structural shortcomings of the model (P9L30-P10L4), and a clarification that this will be the main focus of future work (29L23-29).*
>
> *With regard to including an analysis of speciated mass concentrations, we found that few of the campaigns included both number size distributions as well as speciated mass concentrations (and some included the latter and not the former), complicating the analysis and interpretation. We also felt that the focus of this paper should be on the number distributions, although we use the AMS data where available and appropriate to highlight specific biases. Nevertheless, we have added a sentence to the conclusions highlighting this as a potential avenue for further work (P29L29).*

3. Page 8, line 11: The statement that "interannual variability in aerosol burden is small" may be true, but the long-term trend is not and given that measurements from GASSP

extend almost 2 decades, this statement gives a false impression that variations in emissions are not relevant to these comparisons. This is an inherent weakness of this study (comparisons with only one model year), and in absence of a more detailed analysis of how meteorological and emissions variations over two decades contributes to model-measurement airborne point comparisons, the authors must acknowledge that their comparison is not "apples-to-apples".

> *Indeed, this is a weakness in our study, mainly due to the computational limitations of the large number of sensitivity runs performed. This is now explicitly acknowledged (P12L3-8) and highlighted as another aspect of future work (P29L23-29).*

4. Similarly, Section 3 explores the temporal sampling aspect of model evaluation, but does not address the fundamental temporal mis-match. It may be more appropriate to evaluate a free-running GCM using a 10 year simulation to capture the role of interannual variation in meteorology and/or emissions, particularly on comparisons with temporally-limited, localized campaigns. Could the authors comment on this?

> *We feel this is also now addressed in the additional discussion in Section 3 (P12L3-8).*

5. Section 3: Could the authors also comment on why they did not simply average the GASSP measurement to the model spatial resolution (in order to not penalize the model for its inability to reproduce sub-grid variability) rather than using 2 minute averages?

> *While both of these approaches would produce similar flight-track averages we feel that averaging along the track provides a number of advantages. Firstly, it avoids the problem of flight tracks not being representative of whole grid-cells when only very few observations are present. It also allows to still interpolate model values to these points, which reduces sampling biases.*

MINOR COMMENTS

1. Section 2.1: It would be helpful if the authors clearly describe how they will use the measurements in Table 1. It appears that they largely focus on the DMA, OPC, and SMPS measurements, with sulfate mass concentrations from the AMS shown in the Appendix.
   > *Good point, this is now described in Section 2.1.*
2. Figure 1: flight tracks are illegible. Make flight tracks finer so that they don't ap- pear as blotches, and consider different colour scheme that enables differentiation of campaigns in the same region.
   > *The color scheme has been carefully chosen to try and discern the (very many) datasets while still being color-blind safe. We've now added hatching to every other dataset to try and make them more distinct and made each flight point smaller to try and make them visible (previously we had intended only to show the broad region of each campaign).*
3. Page 14, line 6: suggest replace "well" with "best" – biases that exceed 50% are not suggestive of a very good simulation. Similarly, the authors should temper their language on page 27, line 6.

> *We've replaced this as suggested and caveated the statement on Page 28 line 7.*

4. Page 14, line 9: should this be 10nm? There is no 20 nm cut-off presented in Figure 4
   > *Yes, thank you.*

5. Page 15, line 18: "near the surface" – suggest replace with "in the boundary layer" as these biases extend through several kms
   > *Gratefully acknowledged*

6. Page 15, line 18: "local emissions sources not resolved by the coarse model resolution" seems an unlikely explanation for discrepancies. Sub-grid plumes may not be resolved by the model, but these should not impact medians. Furthermore, emissions inventories generally do include point source emissions and while specific localized sources may be missing from inventories, they are not de facto globally biased low.
   > *This statement was not very clear. We've updated the text to try and clarify that we mean that the model is unlikely to be able to reproduce the near-source values of number concentrations in emissions plumes sampled by the aircraft due to the resolution (rather than the emissions themselves not being represented).*

7. Page 15, line 25: might insufficient marine organics also contribute to this bias?
   > *This is a possibility, and now noted on P16L34.*

8. Figures 5, 6: it would be useful to include the number of points in these sub-set comparisons on the figures.
   > *Good point, thank you. This are now included in Appendix A*

**Reviewer 3**

Page 2, line 2. The indirect forcing also depends on ice nuclei.
> *Clarified*

Page 2, line 12. Suggest also citing Kacenelenbogen, M., M. A. Vaughan, J. Redemann, R. M. Hoff, R. R. Rogers, R. A. Ferrare, P. B. Russell, C. A. Hostetler, J. W. Hair, and B. N. Holben. 2011. "An Accuracy Assessment of the CALIOP/CALIPSO Version 2/version 3 Daytime Aerosol Extinction Product Based on a Detailed Multi-Sensor, Multi-Platform Case Study." Atmospheric Chemistry and Physics 11 (8): 3981–4000.
> *Added, thank you*

Page 2, line 34. Insert "in situ" between aircraft and measurements.
> *Added*

Page 9. Line 25. omponents should read components.
> *Fixed*

Page 10., line19. VOCALS also did repeated sampling. Cite Wood, R., C. R. Mechoso, C. S. Bretherton, R. A. Weller, B. Huebert, F. Straneo, B. A. Albrecht, et al. 2011. "The VAMOS Ocean-Cloud-Atmosphere-Land Study Regional Experiment (VOCALS-REx): Goals, Platforms, and Field Operations." Atmospheric Chemistry and Physics 11 (2): 627–54.
> *Good point, added*

Page 13, line 1. Clarke and Kapustin (2002) discuss the high number concentrations at >8 km altitudes in the tropics. Refer to it.
> *Done*

Page 15, line 11. Explain why the uncertainty in DMA charging results in erroneous counting but not sizing.

*We've added a short explanation which hopefully makes this clearer.*

Page 17, line 14. "reduced negative bias" is taken to mean an increased number from the model. Explain why the slower condensational ageing shows this for larger particles.

*We've added a hypothesis for this.*

[revised manuscript text omitted]